



# Versatile soil gas concentration and isotope monitoring: optimization and integration of novel soil gas probes with online trace gas detection

Juliana Gil-Loaiza[1], Joseph R. Roscioli[2], Joanne H. Shorter[2], Till H. M. Volkmann[3,4], Wei-Ren Ng[3], Jordan E. Krechmer[2], Laura K. Meredith[1,3,*]

[1] School of Natural Resources and the Environment, University of Arizona, Tucson, AZ, 85721, USA
[2] Aerodyne Research Inc., Billerica, MA, 01821, USA
[3] Biosphere 2, University of Arizona, Oracle, AZ, 85623, USA
[4] Applied Intelligence, Accenture, Kronberg im Taunus, Hesse, 61476, Germany.

*Correspondence to*: Laura K. Meredith laurameredith@email.arizona.edu

**Abstract.** Gas concentrations and isotopic signatures can unveil microbial metabolisms and their responses to environmental changes in soil. Currently, few methods measure soil trace gases such as the products of nitrogen and carbon cycling, or volatile organic compounds (VOCs), that could constrain microbial biochemical processes like nitrification, methanogenesis, respiration, and microbial communication. Versatile trace gas sampling systems that integrate soil probes with sensitive trace gas analyzers could fill this gap with measurements resolving spatial (centimeter scale) and temporal (minutes) variations in concentrations and isotopic signatures of in situ soil gases. We developed a system that integrates new 15 cm long sintered PTFE diffusive soil gas probes with various infrared spectrometers and a VOC mass spectrometer. The system is based on porous and hydrophobic soil probes that non-disruptively collect and transport gas from multiple probes to one or more central gas analyzers. Here, we demonstrate the feasibility and versatility of an automated multi-probe system for soil gas measurements of isotopic ratios of nitrous oxide ($\delta^{18}O$, $\delta^{15}N$, and the $^{15}N$ site-preference of $N_2O$), methane, carbon dioxide ($\delta^{13}C$), and VOCs. First, we used an inert silica matrix to challenge probe measurements under controlled gas conditions. By changing and controlling system flow parameters, including probe flow rate, we optimized recovery of representative soil gas samples while reducing sampling artifacts on subsurface concentrations. Second, we forced environmental manipulations in soil-filled columns to demonstrate real time detection of subsurface gas dynamics in response to irrigation and soil redox conditions. In addition, we developed a new laser spectrometer to recover isotope ratios for $^{14}N^{14}N^{16}O$ ("$\delta446$"), $^{14}N^{15}N^{16}O$ ("$\delta456$"), $^{15}N^{14}N^{16}O$ ("$\delta546$"), and $^{14}N^{14}N^{18}O$ ("$\delta448$") with high precision and low concentration dependence. We captured temporal subsurface gas pulses in $CO_2$, $N_2O$, and VOCs. This demonstrated the potential for diffusive-based probes to couple to trace gas sensors for soil health and fertility studies, and to inform high-throughput meta-omics, leading to the development of a suite of powerful new tools for soil analysis.



## 1 Introduction

The impact of soil gas fluxes on atmospheric composition is typically measured at the soil surface, yet new belowground approaches may provide a more mechanistic perspective of trace gas cycling. Soil is a source and sink of important trace gases including nitrous oxide ($N_2O$), carbon dioxide ($CO_2$), methane ($CH_4$), and volatile organic compounds (VOCs). Soil fluxes are driven by abiotic and biotic processes including microbial metabolism, and soil environmental conditions (Conrad, 2005; Jiao et al., 2018; Karbin et al., 2015), both of which vary in space (i.e. soil aggregate (Schimel, 2018) to field (Wang et al., 2014) and time (e.g. rain-driven $N_2O$ emission pulses). For example, carbon and nitrogen cycling processes that produce $CO_2$ (respiration), $CH_4$ (methanogenesis), $N_2O$ (denitrification), and volatile organic compounds (VOCs) are affected by variables including temperature, oxygen level, soil moisture, and nutrient availability that can change on a fine spatial scale in a short period of time (Jiao et al., 2018). Soil moisture and oxygen modulate denitrification processes in soil, and hence emissions of $N_2O$ (Groffman et al., 2009) and VOCs (e.g. butane, benzene, and methanol (Abis et al., 2020; Raza et al., 2017)). Soil conditions also affect $CH_4$ production and consumption from microbial communities at micron to centimeter scales (Schimel, 2018). These spatial and temporal variations in belowground processes strongly influence subsurface gas dynamics, yet constraining how this belowground variability affects soil fluxes has been limited by an inability to make real time and in situ measurements. As a result, the contribution of processes and influence of drivers to net surface fluxes remains buried.

In addition to contributing to atmospheric composition via above-ground fluxes, soil gases serve as messengers of belowground biogeochemical processes and microbial activity. Soil microbial metabolism produces specific trace gases via biochemical pathways that impart characteristic isotopic signatures to trace gas moieties. Measurements of stable isotopomers (abundance and position) of soil trace gases can therefore be a valuable tool to identify and quantify gas processes (Yoshida and Toyoda, 2000). For example, the $CH_4$ production pathway (acetoclastic, hydrogenotrophic, or methylotrophic methanogenesis) can be identified by the ratio of rare $^{13}CH_4$ to the abundant $^{12}CH_4$ (McCalley et al., 2014; Penger et al., 2012)). Similarly, the ratio of $^{15}N$ to $^{14}N$, and the position of the $^{15}N$ relative to the O in $N_2O$ depends upon its production pathway, such as hydroxylamine decomposition, chemodenitrification, nitrifier denitrification, or denitrifier denitrification (Sutka et al., 2006; Yoshida and Toyoda, 2000). Measurements of all three isotopic properties of $N_2O$ ($^{15}N$ abundance, $^{15}N$ site preference, and $^{18}O$ abundance) can identify the type of biochemical process generating the $N_2O$, and the microbe type (bacterial, archaeal, or fungal) (Toyoda et al., 2017). VOCs are signals for diverse microbial and chemical interactions in soils. They are metabolites and signaling molecules involved in microbial and plant-microbe interactions such as quorum sensing, and they can reflect soil health, stress responses, and microbial identity (Insam and Seewald, 2010; Schulz-Bohm et al., 2018). Advances in tracking soil microbial activity using trace gas messengers will elevate the understanding of the role of microbial communities and their metabolism in soil health, greenhouse gas cycling, and biological interactions in soil.

Over the years, soil gas sampling approaches have evolved to recover gas samples with less disruption to the soil environment. Early soil gas sampling methods inserted rigid perforated tubes or wells into the soil to withdraw soil gas by





suction using a syringe (Holter, 1990), pump (Maier et al., 2012), or other manual sampling method (Panikov et al., 2007).
This methodology to extract gas from the soil pores was time consuming and created artifacts by driving advective flow,
disturbing the probe surroundings, transporting gas beyond the probe area compromising the real gas concentration (Maier et
al., 2012), and in some cases, withdrawing insufficient sample volume for analysis. To overcome the drawbacks of advective
gas sampling, diffusive probes sample soil gases by non-advective gas exchange driven by molecular diffusion across a
diffusive membrane. Porous membranes allow the partitioning of the gas from soil and liquid phases (Volkmann et al., 2016a,
2016b). One drawback of diffusive sampling probes has been the relatively large volume needed to generate sufficient sample
for gas analyzers and the correspondingly long time required for the internal sampling volume to reach equilibration with soil
gas. For example, probes larger than 1 m have been used in water (Rothfuss et al., 2013) and soil (Jacinthe and Dick, 1996),
and small silicone probes have required extended sampling times (>7-48 hours) to equilibrate (Kammann et al., 2001)
(Petersen, 2014). Polypropylene (Accurel, V8/2HF, Membrana GmbH, Germany) materials improved equilibrium time at an
equivalent probe length (Flechard et al., 2007; Gut et al., 1998; Rothfuss et al., 2013). Long probes disturb soil, especially
upon installation, spurring the interest in discovering new materials that enhance diffusion at a smaller probe size while still
resolving gas concentrations and isotopic signatures. For example high density materials like expanded polytetrafluoroethylene
(PTFE), and polyethylene promote better equilibration time than silicone (DeSutter et al., 2006) increasing temporal resolution
from hours to minutes in different matrices, e.g. water isotopes in soil (Volkmann and Weiler, 2014) and tree xylem (Volkmann
et al., 2016a), and $CO_2$ in soil (DeSutter et al., 2006). These materials recovered representative gas concentrations and isotopic
signatures, but have been limited by cracking, water infiltration (Volkmann et al., 2016a, 2016b), and soil disruption during
sampling (Hirsch et al., 2004). Nonetheless, diffusive sampling approach is a promising means for non-destructively
recovering soil gas for analysis. The search is ongoing for materials that equilibrate efficiently by diffusion with minimal probe
length.

Probes face multiple demands in the soil system during field deployment. For long-time monitoring in the field,
subsurface probes must be robust to extreme weather, plant and microbial activity, and other site disruptions that could affect
the integrity of the porous membrane. The probe material must tolerate environmental changes and interactions with biotic and
abiotic factors. Microbial interactions with probe materials can reduce probe integrity (Rothfuss and Conrad, 1994), modify
gas concentrations, or reduce gas exchange by biofouling (Krämer and Conrad, 1993). Small soil particles can clog probe pores
and limit gas diffusion and probes can also break or crack in freeze-thaw cycles (Burton and Beauchamp, 1994; Gut et al.,
1998) or during installation (Volkmann et al., 2016a, 2016b). Probe membranes must resist water break-through, which has
caused water interference problems in nylon (Burton and Beauchamp, 1994) and polypropylene (Gut et al., 1998) probes. The
limitations of some probe materials have been evaluated under controlled conditions (DeSutter et al., 2006; Munksgaard et al.,
2011; Rothfuss et al., 2013). To address these challenges, new non-reactive and hydrophobic porous probe material is needed
to meet the demands of long-term soil sampling.

An advantage of diffusive soil gas probes is that they can be integrated with online soil gas sampling instrumentation
to quantify soil gas concentrations and isotopic signatures (Gangi et al., 2015; Gut et al., 1998; Rothfuss et al., 2013; Volkmann





et al., 2016b, 2018). So far, a fraction of available trace gas analyzers have been integrated with online (e.g. $H_2O$, $CO_2$, $CH_4$)
diffusive soil gas sampling systems, leaving the majority of gas cycling and microbial messengers untapped. Tunable Infrared
Laser Direct Absorption Spectrometers (TILDAS) measure important small molecules such as $N_2O$, $CH_2$, NO, $CO_2$, and CO.
VOCs, valuable markers of different and highly specific biogeochemical processes, can be monitored with Proton Transfer
Reaction Time Of Flight Mass Spectrometers (PTR-TOF-MS). There is a need to test sampling systems under controlled
conditions with promising new probe materials to integrate with the diverse suite of available high precision gas analyzers.
The methods should be optimized for different gases, recognizing differences in molecular diffusivity (exchange across probe)
and surface interactions (partitioning to tubing). Online, multiplexed systems have been deployed in the field to measure
multiple spatial points (Jochheim et al., 2018; Volkmann and Weiler, 2014). Expanding the gases that can be sampled by
diffusive soil gas sampling probes in the field will increase understanding of biogeochemical processes through measurements
at a refined temporal and spatial resolution.

In this study, we describe a real time soil trace gas sampling system that integrates diffusive soil probes with TILDAS

and PTR-TOF-MS Vocus analyzers to capture fast, spatially resolved production and isotopic signatures of key soil trace gases
and their responses to environmental changes. To achieve this, we developed diffusive, hydrophobic soil probes from sintered
PTFE (sPTFE) and used custom controlled soil columns to evaluate their ability to retrieve gas samples via continuous
sampling. We measured trace gas messengers of microbial nitrogen and carbon cycling ($N_2O$, $CH_4$, CO, $CO_2$,,VOCs), including
the site-specific stable isotopologues of $N_2O$ and $CH_4$. The sPTFE probe material has uniform pore distribution, improving gas
diffusion, and is chemically and biologically inert and terminally resistant, properties that make this material a good candidate
for long term soil gas probes. We evaluated the integration of the probe and the analyzers using columns with (i) a silica matrix
with controlled gas concentrations, and (ii) a soil matrix with manipulated environmental conditions, mimicking precipitation
and changing redox conditions of the soil. In addition, we optimized the TILDAS sample cell volume, sample transfer schemes
and flow rates, and the instrument's concentration dependence. Here, we show that improving soil gas sampling methods and
coupling to online high-resolution instrumentation can lead to a robust and flexible system that measures a wide array of soil
trace gases and will give a snapshot of microbial activity and biogeochemical cycles in soils.



## 2. Materials and Methods

### 2.1 Probes and prove evaluation system

### 2.1.1 Sintered PTFE (sPTFE) probes

We built gas permeable soil probes from microporous tubes of sPTFE (Fig. 1a). sPTFE is hydrophobic, allowing gas diffusion across the tube while preventing liquid water breakthrough. The material is structurally stable and non-reactive making it suitable for long-term use in soils. We selected a total of four probes with different pore sizes and tube dimensions including the outer diameter, inner diameter, and wall thickness (Table 1) to evaluate their equilibration properties and soil impacts. Probes were machined (White Industries, Inc., Petaluma, CA) from solid sPTFE blocks (Berghof GmbH, Eningen, Germany). In some cases probes were made from two pieces (Table 1) assembled as a single probe using perfluoroalkoxy (PFA) unions (Swagelok, Solon, OH). We constructed probe prototype assemblies to connect probes to inlet and outlet transport lines of 1/8" fluorinated ethylene propylene (FEP, Versilion™, Saint-Gobain, Malvern, PA) using stainless steel reducing unions (Swagelok, Solon, OH). In some cases, PTFE tape was used to account for mismatch between fittings and probe outer diameters. After assembly, each probe was submerged under water while flowing ultra-zero air gas through the probe test for leaks in the fitting assembly. This was to ensure that probe sample concentrations were the result only of gas diffusion across the porous wall of the probe.

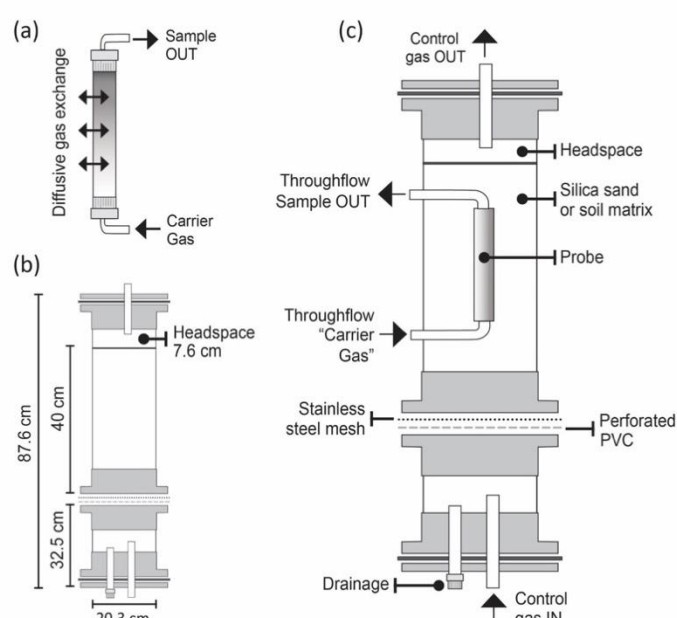

**Figure 1.** Lab-based custom soil column assembly built to evaluate probe performance against controlled gas concentrations in a silica sand and soil matrix. (a) microporous probe of sPTFE, (b) dimensions of the two column sections, and (c) column components to enable controlled probe evaluation.





**Table 1**. List of sPTFE probe pore size and dimensions including outer diameter (OD), inner diameter (ID), and wall thickness (W)

| Probe ID (pore size in μm) | Dimensions (mm) (OD x ID x W) | Length (mm) |
|---|---|---|
| P5 (5) | 12.7 x 6.3 x 1.6 | 147.5 |
| P8 (8) | 12.7 x 6.3 x 1.6 | 147.5 |
| P10 (10) | 12.7 x 6.3 x 1.6 | 147.5 |
| P25* (25) | 9.5 x 4.7 x 2.4 | 147.5 |

* Two sPTFE pieces joined with a PFA fitting

**2.1.2 Soil Columns**

We built experimental columns to (1) evaluate probe performance under controlled gas supply and in a non-reactive matrix (silica sand), and (2) measure in situ gas dynamics in response to environmental manipulations (e.g. wetting, redox state) in a complex matrix (soil). Our custom soil columns allowed a gas of controlled composition (control gas) to be advectively forced through the matrix from below (Fig. 1). We placed silica sand in the columns and continuously flushed them with known gas concentrations and isotope ratios to evaluate probe performance (System 1 tests at University of Arizona; UA, and System 2 tests at Aerodyne Research Inc., ARI, Section 2.3.1). We also used the columns to drive controlled environmental manipulations including controlled wetting and forced shifts from anaerobic to aerobic conditions to measure in situ gas dynamics in complex soil (System 2 tests at ARI, Section 2.3.2).

Each column consisted of two sections (Fig. 1b): a lower section (32.5 cm) supporting drainage and buffered delivery of control gas, and an upper section (40 cm) containing the matrix (silica or soil), with a headspace layer for uniform column outflow (7.6 cm). Together, the two column sections were 20.3 cm inner diameter, 87.6 cm length (including base and cover), 28 L volume and constructed from schedule 80 PVC. The central location of the probe in the upper section allowed sufficient distance from column walls (10 cm) and the soil/gas interface (15.2 cm) to avoid edge effects (Fig. 1c). The upper and lower column sections were separated by a layer of perforated PVC (staggered 1/8 in. holes and 40% open area) and a type 304 stainless steel wire cloth mesh (325 x 325 mesh (44 μm), 0.051 mm opening size) to allow passage of control gas and drainage of water, while retaining matrix integrity in the upper section. During controlled irrigation experiments, excess water in the matrix dripped into the lower section and out a drain on the base (sealed during sampling). Column sections were joined using schedule-80 PVC flanges, bolts, and rubber gasket seals allowing columns to be modular and easy to disassemble, transport,





and refill. Additionally, PTFE and polyetheretherketone (PEEK) bulkhead fittings (IDEX Health & Science LLC., Oak Harbor,
WA, USA) and washers provided air-and water tight connections for the sampling and control gas tubing. The gas permeable
soil probes were installed in the center of the column, and can be flanked by soil sensors (e.g. moisture, temperature) (Fig.
1c).

Silica sand (Granusil 4095; high purity industrial quartz; Covia Corporation, Emmett, Idaho) was used as the non-

reactive matrix to evaluate the effect of probe sampling on the matrix and probe performance. This silica is a non-reactive low
alkaline oxide matrix with a characterized particle size distribution (Table S1), thus allowing absolute concentration
measurements of the control gases
**2.1.3 Gas sampling system**

The soil probe sampling system operated in a continuous flow mode. In this approach, for each sample measurement

we flowed carrier gas through the soil probe to equilibrate with soil gas (probe flow), then diluted the outflow online (dilution
flow) and sent the combined flow (total flow) to the gas analyzer for real time measurement. The gas sampling system consisted
of the following components: custom gas control, probe sampling, and a measurement and data acquisition system that
integrated three gas columns (Fig. 2). Similar sampling systems were built at UA (System 1) and Aerodyne (System 2) and
differed in the specific TILDAS and gas control components deployed at each location (Table 2). In order to prevent bulk gas
advection in the soil it was critical to ensure that flow into and out of the probe were matched such that probe + dilution =
instrument intake. As such, system control and sampling depended on precise flow control by digital mass flow controllers
(MFC, Alicat Scientific, Tucson, AZ, USA). Dilution flow was important in the system (Fig. 2) to reduce risk of condensation,
avoid exceeding optimal detection range, and increase gas analyzer cell response time. The control gas system allowed us to
stipulate the specific mole fractions and relative isotope mixtures at the column inlet. Custom control gas composition was
mixed from Ultra Zero Air (UZA; Airgas Inc.) and concentrated gas cylinders (e.g. 5% $CO_2$; Table 3). A bypass line was
installed to independently verify the control gas composition entering the column while the column outflow line was used to
measure column headspace concentrations (Fig. 2). A needle valve was added to the bypass line in System 2 to address
inconsistent dilution rates observed in the control gas bypass line of System 1 (Fig. 2). We used UZA as the probe sampling
carrier gas. Two streams of UZA controlled by MFCs were delivered in tandem through a stream selector 16x2 port valve
(VICI Valco Instruments Inc. Houston, TX, USA) to the probe inlet (probe flow) and a PEEK "tee" connection at the probe
outlet for dilution flow. The latter diluted the probe gas stream, with the total flow directed to the analyzer (Fig. 2). A second
stream multiport selector (VICI Valco Instruments Inc. Houston, TX, USA) was used to select the resultant total flow to deliver
to the analyzers.

We integrated multiple columns with the TILDAS using an automated multi-valve system. In System 1, we used a

custom LabVIEW (National Instruments, Austin, TX) program to precisely execute scripts generated in Matlab (The
MathWorks Inc.; 2018. Natick, Massachusetts) for timing and control of MFC gas flow rates and VICI valve switching. The
LabVIEW program queried and logged MFC parameters and SDI-12 via USB multi-drop box (BB9-RS232, Alicat Scientific,





Tucson, AZ, USA) interfaces. In System 2, TDLWintel, the TILDAS measurement and data acquisition program, controlled
the multi-valves on a schedule for continuous unattended operation.

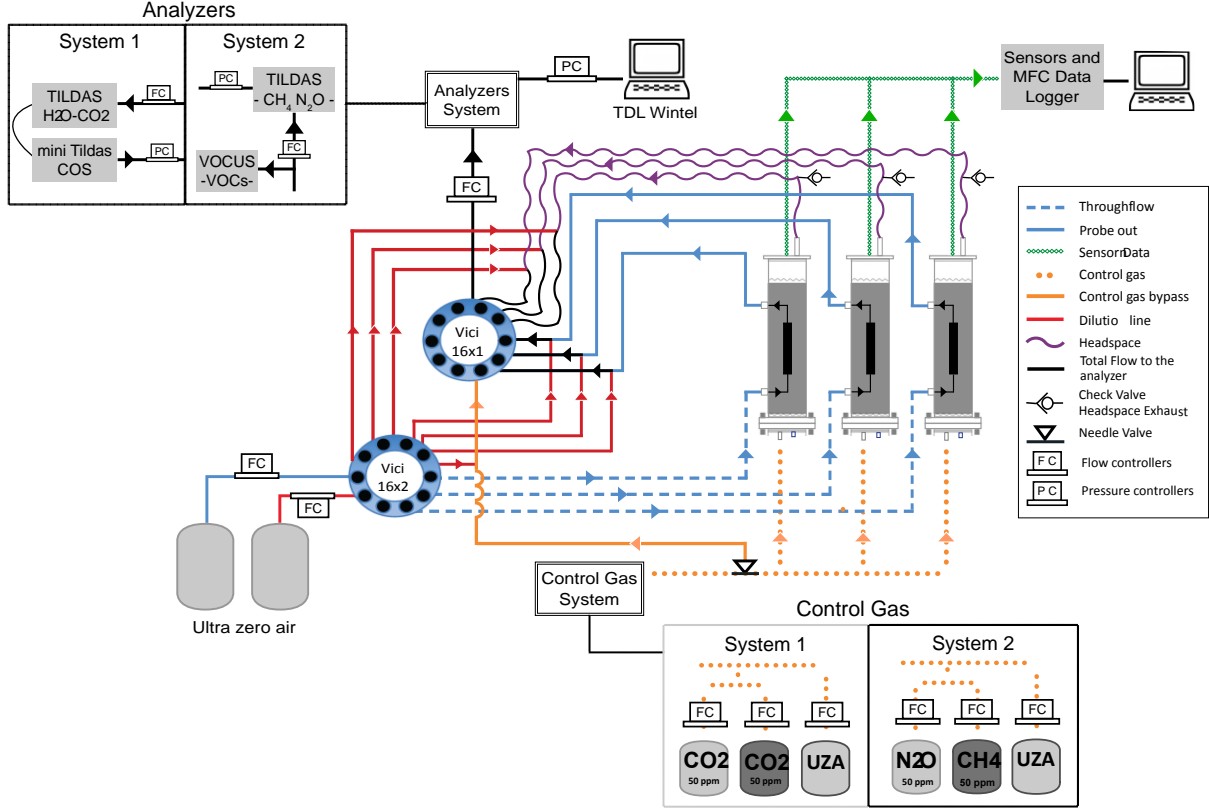

**Figure 2.** Detailed schematic of sampling System 1 (UA) and System 2 (ARI). Column matrix gas concentrations were
controlled by mixing cylinder gas with UZA using MFCs and delivering the custom gas mixture through the columns from
bottom to top (orange dotted line). Probe sampling flow rates were controlled precisely using three MFCs to ensure that flow
in and out of the probe was balanced (*sample flow* (blue lines) + *dilution flow* (red lines) = *total flow to analyzer* (black lines)).
Column headspace (atmospheric pressure) and control gas bypass (positive pressure) were controlled by MFCs at two points
(*dilution, total flow to analyzer*), forcing the *sample flow* as a makeup flow (*sample flow = total flow – dilution flow*).



**Table 2.** Contrasting features between Systems 1 and 2

| Feature | System 1 | System 2 |
|---|---|---|
| Objective | Feasibility of probe-TILDAS integration | Versatility of soil gas probe sampling |
| Location | Biosphere 2, University of Arizona, Tucson, AZ | Aerodyne Research Inc., Billerica, MA |
| Analyzer 1 | Dual-laser TILDAS for $H_2O$ and $CO_2$ isotopes | Novel dual-laser TILDAS for $N_2O$ and $CH_4$ isotopes |
| Analyzer 2 | Mini TILDAS for OCS, CO, $CO_2$, and $H_2O$ | Vocus PTR-T OF-MS for VOCs |
| Control Gas (bulk) | Ultra-Zero Air | Ultra-Zero Air; Ultra-High Purity $N_2$ |
| Control Gas (trace) | 5% $CO_2$ in air | 49.1 ppm $N_2O$ in air; 54.6 ppm $CH_4$ in air |
| Flow Control | 0.6 to 1 SLPM per column | 0.65 SLPM per column |
| Matrix | Silica | Silica, Soil |


To evaluate the probe and the column performance, we corrected observed concentrations ($C_{obs}$) using the ratio of the
dilution and total flows to obtain true probe sample, column/headspace, and control gas concentrations (C). For example Eq
(1), for soil probe sample concentrations we used the ratio of the total flow ($F_t$; probe sample plus dilution flow) to the probe
sample flow ($F_p$):
$C = C_{obs} * F_t / F_p$                                                                                     (1)
**2.2 Trace gas analyzers**
To characterize soil probe performance, we used a suite of trace gas analyzers relevant to biological soil gas cycling
(Fig. 2). The system and analyzers were modified as needed to integrate with the soil probe sampling system. TILDAS isotope
analyzers measure the concentrations of individual isotopologues, and isotopic ratios can be determined using Eq. (2),
$\delta^i X = ( R_n / R_{reference} - 1) \times 1000$                                                          (2)
where, $R_n$ refers to the ratio of the rare isotopomer, $^iX$, to its abundant isotopomer (Toyoda et al., 2017).



### 2.2.1 Coupled laser spectrometers for CO₂ and H₂O isotopes and COS and CO


System 1 integrated two TILDAS trace gas analyzers (Aerodyne Research, Inc., Billerica, MA, USA) with the soil
probe testing system to evaluate the feasibility of coupling the sintered PTFE probes with analyzers and evaluate performance
under controlled conditions. TILDAS-1 was a dual-laser instrument configured for measurement of $^{1}H^{16}O^{1}H$, $^{1}H^{16}O^{2}H$, and
$^{1}H^{18}O^{1}H$ at 3765 cm$^{-1}$ and $^{12}C^{16}O^{16}O$, $^{13}C^{16}O^{16}O$, $^{12}C^{16}O^{17}O$, $^{12}C^{16}O^{18}O$ O at 2310 cm$^{-1}$ with a 18 m absorption cell. TILDAS-
2 was a compact single-laser instrument configured to quantify carbonyl sulfide (OCS), carbon monoxide (CO), water (H₂O),
and CO₂ at 2050.4-2051.3 cm$^{-1}$ with a 76 m absorption cell. The dual and Mini TILDAS analyzers had a 500 cm³ and 300 cm³
sample cell volume, respectively. The platform draws air samples through an absorption cell at low pressure where laser light
is transmitted in a multi-pass configuration for long effective absorption path lengths. The laser is scanned at kilohertz rates
over the rovibrational absorptions of the molecule(s) of interest. Transient light absorptions were fit to known Voigt profiles to
determine molecular concentrations on-the-fly using Aerodyne's proprietary acquisition and analysis software, TDLWintel.
For this experiment we connected the two TILDAS analyzers at controlled flow rate (500-250 sccm, MC-1SLPM-D, Alicat)
in series, and cell pressure was dynamically controlled to 40 Torr (PCSC-EXTSEN-D-15C/5P, Alicat) between the two
analyzer sample cells and vacuum pump (MPU2134-N920-2.08, KNF Neuberger, Trenton, NJ). The TILDAS optical tables
were each purged with 100 sccm zero air.
In System 1, CO₂ concentrations varied linearly with controlled dilutions of 10% CO₂ tanks (Fig. S1 dual CO₂ cal),
and absolute CO₂ concentrations were calibrated with a linear curve. We calibrated the $\delta^{13}C$-CO₂ from the concentration
dependent relationship of $\delta^{13}C$-CO₂ vs observed [CO₂] (Fig. S2); specifically, we fit a gaussian equation to the relationship
between ($\delta^{13}C$-CO₂ observed - $\delta^{13}C$-CO₂ true ~ -39.2 ‰ vs Vienna PeeDee Belemnite (VPDB)) and CO₂ concentration
(accounting for standard deviation in $\delta^{13}C$-CO₂ measurements). We applied this CO₂-dependent correction to all $\delta^{13}C$-CO₂
values reported here.

### 2.2.2 Novel laser spectrometer for N₂O and CH₄ isotopomers


System 2 integrated a second and nearly identical (Table 2) gas sampling system with a novel dual TILDAS analyzer
for isotopomers of methane (CH₄) and nitrous oxide (NO₂) (Aerodyne Research, Inc., Billerica, MA, USA) to test instrument
modifications that help integrate soil gas sampling probes with laser spectrometry and demonstrate the versatility of soil probe
gas sampling.
In this study, we identified and selected the best spectral region and laser technology for continuous high precision
measurements of isotopomers of CH₄ ($^{12}CH_4$ and $^{13}CH_4$), and N₂O ($^{14}N^{14}N^{16}O$ ("446"), $^{14}N^{15}N^{16}O$ ("456"), $^{15}N^{14}N^{16}O$ ("546"),
and $^{14}N^{14}N^{18}O$ ("448")). The regions near 2196 cm$^{-1}$ (4.56 μm) and 1295 cm$^{-1}$ (7.72 μm) provide interference-free
measurements of N₂O and CH₄, respectively, and their rare isotopes. The 2196 cm$^{-1}$ region is also capable of measuring CO₂
at soil-relevant concentrations (parts-per-thousand levels). The CH₄ and N₂O TILDAS system was optimized with respect to
optical alignment, laser operating parameters (i.e. scan length, laser current and temperature settings), and fit parameters.





Short- (seconds) and long-term (minutes-hours) noise were determined by sampling from a compressed air cylinder as a
constant gas source, followed by Allan-Werle variance analysis (Werle et al., 1993). We chose 30 Torr as the optimum cell
pressure to minimize both noise and spectral crosstalk between isotopomer absorptions. To reduce sample volume we designed
a new cell insert and a compact 76 m pathlength multipass sampling cell. The novel volume-reducing insert for the 76 m cell
has interior walls that match the contour of the multipass pattern and was 3D-printed using PA2200 nylon. After printing, the
interior and exterior surfaces of the insert were sealed with urushi lacquer—a stable, durable, inert lacquer (McSharry et al.,
2007). The turnover time of the cell volume with insert was evaluated in continuous sampling mode.
The concentration dependence of isotope δ values derived from infrared isotopic measurements is an analytical
challenge that is instrument dependent. To minimize the concentration dependence we used: (i) frequent backgrounds to
minimize offsets (i.e. immediately prior to each sample measurement), and (ii) identified best fitting parameters for each
spectral region and application. During System 2 operation, we automated script schedules using an external command
language (ECL) within TDLWintel that ran backgrounds, calibrations, and controlled valves.
Alcohols (e.g. methanol and ethanol) have weak features in the methane spectral window (1295 cm$^{-1}$), at levels
typically below that of the isotopic precision. We tested whether VOCs would cause infrared spectral interferences with
TILDAS analysis by exposing the instrument to artificially elevated part-per-thousand levels of methanol, ethanol, and
formaldehyde—three species that may be common in soil. We found potential for interference near the $^{13}CH_4$ absorption at
elevated alcohol levels, but did not observe this interference in the spectra collected from probes in the soil tested.
The System 2 calibration system used online mass flow control to dilute concentrated $N_2O$ or $CH_4$ calibration gases
into UZA. We used pure samples of $N_2O$ from Massachusetts Institute of Technology (MIT Ref I and Ref II). The isotopic
ratios of $N_2O$ were determined by Isotope Ratio Mass Spectrometry (IRMS) and TILDAS measurements, and externally
verified by *S. Toyoda* at Tokyo Institute of Technology (McClellan, 2018). For calibration of the soil matrix tests discussed
below, we used MIT Ref II to make a surveillance standard of 1,000 ppm $N_2O$. After calibrating $N_2O$ isotopes against the
reference gas, observed lab air $N_2O$ isotopic ratios were within 3‰ of the relatively stable isotopic ratios of ambient
tropospheric $N_2O$ (Snider et al., 2015): bulk $^{15}N$ value of 6.3-6.7‰, and site preference of 18.7‰ (Mohn et al., 2014), and $^{18}O$
value of 44.4‰ (Snider et al., 2015). For $CH_4$ concentrations, a $CH_4$ surveillance tank served as a stable isotopic source to
identify changes in isotopic composition.
**2.2.3 High resolution volatile organic compound gas analyzer**
In System 2 experiments, we integrated a Vocus PTR-TOF-MS (Aerodyne Research Inc., Billerica, MA, USA)
(Krechmer et al., 2018) into the sampling system in parallel with the $N_2O$/$CH_4$ TILDAS, to detect soil VOCs such as
monoterpenes, isoprene, and pyruvic acid (Gonzalez-Meler et al. 2014; Guenther et al. 1995). The Vocus technology contains
a corona discharge reagent-ion source and focusing ion molecule reactor (fIMR) that has low limits of detection (less than part
per trillion by volume) and fast time response, acquiring the entire mass-to-charge spectrum on the order of microseconds. A





TOF instrument also has high resolving power in the mass dimension, enabling separation of isobaric signals (occurring at the
same nominal mass-to-charge ratio). The TOF employed in this work consisted of a 1.2 m flight tube enabling a resolving
power > 10000 m/dm. A sample flow of 100 SCCM was injected continuously into the Vocus source, with no extra overblow
or carrier flow in the inlet line.
Data was processed using the Tofware (Aerodyne/TOFWERK A.G.) software package in Igor Pro (Wavemetrics).
For these experiments PTR-TOF-MS was not quantitatively calibrated for the signals reported below, as we were only
interested in relative concentration responses to wetting. Thus, signals are reported in non-normalized counts/s (Hz).
**2.3 Experiments performed**
We performed experiments using Systems 1 and 2 (Section 2.2; Fig. 2) to demonstrate the feasibility and versatility
in coupling the permeable soil gas probes to trace gas analyzers to measure in situ gas concentrations and isotope ratios in
soils. We conducted two categories of experiments: 1) *Experiments under controlled conditions using silica*, characterizing
the ability of probe sampling to measure known, controlled soil gas concentrations; and 2) *Experiments with soil,* characterizing
the ability of probes to capture soil microbial gas cycling dynamics from natural soils in response to environmental changes.
**2.3.1 Experiments under controlled conditions using silica**
Silica sand was used to limit trace gas production or consumption from the matrix for controlled evaluation of the
probe. Three columns with one soil gas probe each were filled with dry silica matrix (Table S1) and closed hermetically. Gas
concentrations and isotopic signatures of the inlet, soil probe, and column headspace samples were quantified while the gases
flowed continuously through the column and dilutions rates were varied (Table 3).
We evaluated the *effect of probe sampling on the column* (Experiment 1) by changing the probe flow rate with constant
control gas concentration and dilution. With System 1 and a single column with silica matrix, we alternated measurement of
$CO_2$ concentration in headspace gas (1 h) and the probe (15 min) to determine the impact of probe sampling on soil column
outflow concentrations. Next, we tested the flow conditions that support the probe delivering fully equilibrated and
representative samples by *varying flow and dilution* at constant column concentrations (Experiment 2). We evaluated 42
combinations of set points for total flow (from 50 to 300 sccm, at 50 sccm intervals) and dilution (from 90% to 9%, at 15%
intervals). Each measurement cycle lasted 25 min (15 min probe; 10 min column headspace) using one probe in System 1 and
System 2.
We scaled-up the sampling systems to 3 probes to evaluate multiple online probe sampling (Experiment 3). We
measured probe and headspace gas at a constant dilution (75%) of a 2000 ppm $CO_2$ control gas for a target observation
concentration of 500 ppm and probe flow rates of 5, 10, 20, 30, 40, 50, and 100 sccm (System 1). System 2 was similarly
evaluated with $N_2O$ and $CH_4$ control gases in the silica matrix (Table 3).





**Table 3.** Experiments under controlled conditions with silica matrix using Systems 1 and 2

| Experiment | Columns | Probe Pore Size (µm) | Total flow (sccm); Probe Flow (sccm); Dilution (%) | Control gas (ppm) | System |
|---|---|---|---|---|---|
| 1. Effect of probe sampling (silica)[a] | 1 | P8 (8 um) | total (10-600); probe (5-300); dilution (50%) | $CO_2$ 1000 | 1 |
| 2. Flow and dilution[a] | 1 | P8 (8 um) | total (50:50:300); probe (0-300); dilution (90:15:0%) | $CO_2$ 1000 | 1, 2 |
| 3. Multi-probe evaluation[a] | 1 | P8 (8 um) | total (20-400); probe (5-100); dilution (75%) | $CO_2$ 2000 | 1 |
| | 2 | P10 (10 um) | | | |
| | 3 | P5 (5 um) | | | |
| | 4 | P8 (8 um) | total (250); probe (25); dilution (90%) | $N_2O$ 3ppm $CH_4$ 7 ppm | 2 |
| | 5 | P10 (10 um) | | | |

[a] Experiments 1-3 were conducted with the column top closed and no water addition.
**2.3.2 Experiments with soil**

We replaced the silica matrix with soil in the columns to understand probe behavior and response when monitoring
soil gases in a complex and dynamic soil matrix. We measured $N_2O$ and $CH_4$ concentrations and isotopic signatures with the
improved TILDAS instrument on System 2 (Fig. 2) in a series of experiments (Table 4). For soil experiments, headspace
measurements can be used to track surface gas fluxes, but do not represent control gas concentrations as in the silica
experiments. We evaluated how measured soil gas concentrations changed in response to: probe sample flow rate (Experiment
4); environmental manipulations to the soil matrix (e.g. increased soil moisture with 5.1 cm of simulated rainfall) (Experiment
5); and forced changes to soil redox state (e.g. forced $N_2$ and UZA through the columns to shift from anaerobic to aerobic soil
environments) (Experiment 6). In this last experiment, we integrated the Vocus PTR-TOF-MS to the system to measure soil
VOCs (Fig. 2).



**Table 4.** Experiments under controlled conditions with soil and silica matrix using System 2

| Experiment | Type of soil | Columns | Probe | Total flow (sccm); Probe Flow (sccm); Dilution (%) | Control $N_2O$; $CH_4$ (ppm) | Soil Moisture |
|---|---|---|---|---|---|---|
| 4. Soil vs. silica: multi-probe flow rate dependence | Soil 1 | 4 | P8 (8 um) | total (235); probe (60); dilution (74%) | $N_2O$ 3 ppm; $CH_4$ 7 ppm | Field moisture |
| | Silica | 5 | P10 (10 um) | | | Dry |
| | Silica | 6 | P25 (25 um) | | | Dry |
| 5. Soil wetting[a] | Soil 1 | 4 | P8 (8 um) | total (50-100); probe (25); dilution (50-75%) | | Dry to wet |
| 6. Soil redox: anaerobic ($N_2$) to aerobic (UZA)[ab] | Soil 3 | 5 | P10 (10 um) | total (185); probe (53); dilution (71%) | | Wet |

[a] Experiment conducted with the column top open

[b] Experiment integrated Vocus PTR-TOF-MS for VOCs

**2.4 Data processing**

For System 1, we used RStudio and R version 3.3.2 (Team, 2017) to integrate raw with metadata. Igor Pro (version 7, WaveMetrics, Lake Oswego, OR) for System 1 and System 2 was used to analyze instrument diagnostic, concentrations and times series. We averaged the last 80% to 90% of each measurement. Measurements were dilution corrected to obtain undiluted sample concentrations (Equation 1). In controlled tests when true headspace concentrations were measured before and after a probe measurement, these values were interpolated for comparison against probe concentrations to determine fractional recovery of soil gas concentrations.

**3. Results and Discussion**

**3.1 Instrument improvement ($N_2O$/$CH_4$ isotopomer TILDAS)**

**3.1.1 Selection of spectral regions.**

We selected optimal spectra windows and laser technologies for detection of the isotopomers of both $CH_4$ and $N_2O$ using fundamental rovibrational transitions (Fig. 3). We used Aerodyne-developed simulation programs that utilize the HITRAN database (Rothman et al., 2013) to perform spectral simulations to identify potential measurement regions. Based on these simulations, we obtained appropriate lasers and detectors for the selected spectral regions. Simulations assumed an $N_2O$


mixing ratio of 1 ppm (parts per million, lower end of expected (Rock et al., 2007) in a mixture with 1.3% $H_2O$, 1% $CO_2$, 220
ppb CO and 1.9 ppm $CH_4$, at 30 Torr in a 76.4 m pathlength sample cell. This resulted in the selection of a spectral region
(Fig. 3a) where all four $N_2O$ isotopomers of interest, $^{14}N^{14}N^{16}O$ ("446"), $^{14}N^{15}N^{16}O$ ("456"), $^{15}N^{14}N^{16}O$ ("546"), and $^{14}N^{14}N^{18}O$
("448"), have absorptions in close spectral proximity (<1 cm$^{-1}$), but without overlap of absorptions of each other or other trace
gases such as from $CO_2$. The 2196 cm$^{-1}$ region was used to monitor the $N_2O$ isotopologues and $CO_2$ in the soil gas matrix using
a quantum cascade laser (QCL) (Alpes Laser, Switzerland). We selected a second QCL (Alpes Laser) based on simulations of
methane isotopes in the 1294 cm$^{-1}$ region to monitor $^{12}CH_4$ and $^{13}CH_4$ isotopomers (Fig. 3b). This region also provided
measurement of $H_2O$ content in the soil gas via a water spectral feature at ~1294.0 cm$^1$.

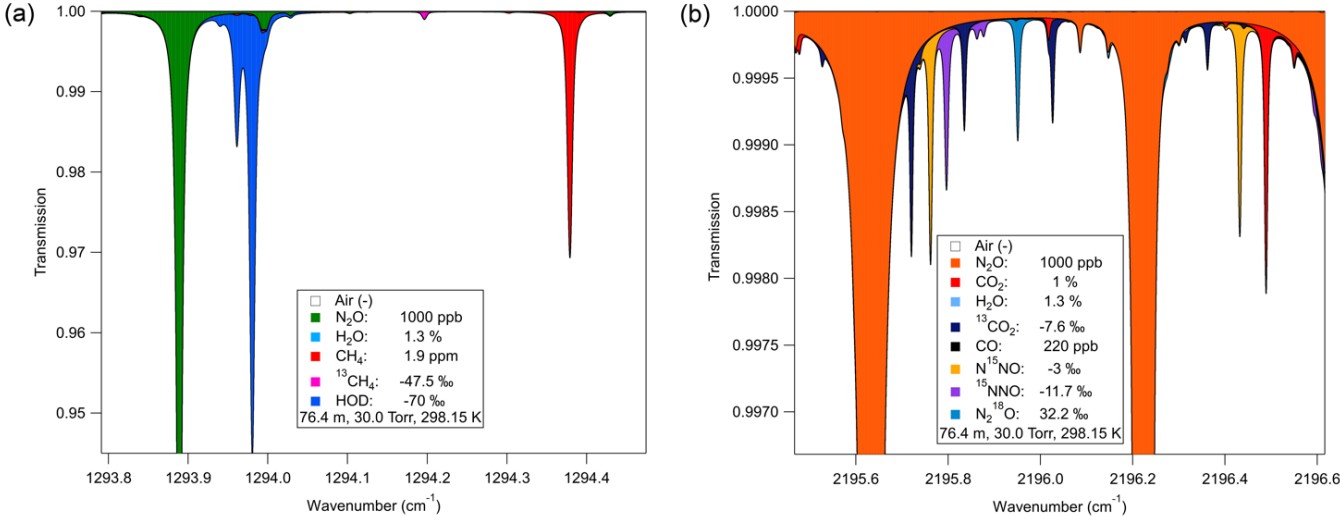

**Figure 3.** Isotopomers spectral regions for monitoring $N_2O$ and $CH_4$ isotopomers. (a) $N_2O$ isotopologue spectrum near 2196
cm$^{-1}$. Four $N_2O$ isotopomers were present and spectrally separated, yellow and purple refer to the $^{15}N$ isotopomers with
different positions relative to the oxygen. Blue refers to the $^{18}O$ isotopomer. (b) Spectral simulation of 1294 cm$^{-1}$ region for
methane analysis with lines well separated from $H_2O$ and $N_2O$.
**3.1.2 Optimization of isotope ratio measurements**
TILDAS operational parameters were optimized to increase isotope ratio precision. For example, we monitored the
slightly weaker doublet at 2196.2 cm$^{-1}$ that had lower concentration dependence than the stronger absorber singlet at 2195.6
cm$^{-1}$ that would produce nonlinear dependence at high mixing ratios. In addition, we modified fitting parameters to minimize
impact of baseline variability on measurement precision (fit shown in Fig. S3). These improvements in spectral fitting helped
minimize the dependency of $N_2O$ and $CH_4$ isotopic ratios on concentration. Specifically, we reduced the slope of $\delta$ vs mole
fraction to 0.7 ‰ ppm$^{-1}$ $N_2O$ (for $N_2O$ < 8 ppm) and 0.5 ‰ ppm$^{-1}$ $CH_4$ (for $CH_4$ < 14 ppm). The online dilution approach was





critical for avoiding $N_2O$ and $CH_4$ concentrations in soil exceeding these linear ranges. We quantified the precision of the
isotopic ratios (Table S2) using Allan-Werle plots (Werle et al., 1993) (Fig. S3).

### 362    3.1.3 Sample cell reduction

We improved measurement response time by reducing TILDAS sample cell volume while maintaining the

spectroscopic path length. Unnecessary 'dead' volume in the sample cell was eliminated through two approaches. First, we
reduced the cell volume (port to port) by 20% (610 $cm^3$ to 485 $cm^3$) by shortening the cell by 4.2 cm, eliminating dead volume
behind the mirrors. Second, the insert reduced the cell volume by ~50% (485 to 245 $cm^3$) by filling volume between the
mirrors, but in the region outside of the multi-pass laser path. Overall, these changes reduced cell volume from 610 $cm^3$
(previous ARI 76-m Astigmatic Multipass Absorption Cell (AMAC) cell) to 245 $cm^3$, which improved the cell response time
by 40%, here defined as the time to observe 75% of a full transition in concentration (Fig. S4) (i.e. from 1.13 (0.005)) s to 0.76
(0.01) s; 30 Torr and 1 SLPM). At the cell pressure of 30 Torr used here, this 245 $cm^3$ absorption cell volume corresponds to
9.7 $cm^3$ of sample gas at ambient pressure.

### 372    3.2 Probe performance under controlled conditions (silica sand)

### 373    3.2.1 Effect of probe sampling on soil gas concentrations (Experiment 1)

Soil probes sample subsurface gases by diffusion across the probe membrane into a UZA stream flowing through the

probe. In our balanced mass flow approach, an equal proportion of UZA molecules diffuse out of the probe relative to soil gas
diffusing in, which can affect (i.e. dilute) concentrations in the subsurface environment. To quantify the impact of probe
sampling on soil column concentrations, we set control gas to 1000 ppm $CO_2$ and varied the probe flow rate from 5 to 300
sccm, and back, at a constant dilution (50%). We evaluated the impact of a 15-min soil probe measurement on subsequent 1-
hour measurements of the column headspace. We found that column $CO_2$ concentrations were depleted directly following
probe sampling (from 0.6 to 1.6% depletion) and took > 1 hour to fully stabilize. Column $CO_2$ was most depleted after higher
probe flow rates (Fig. 4) due to increased $CO_2$-free UZA diffusion through the probe membrane. Low probe flow rates helped
minimize   these sampling artifacts on subsurface concentrations.





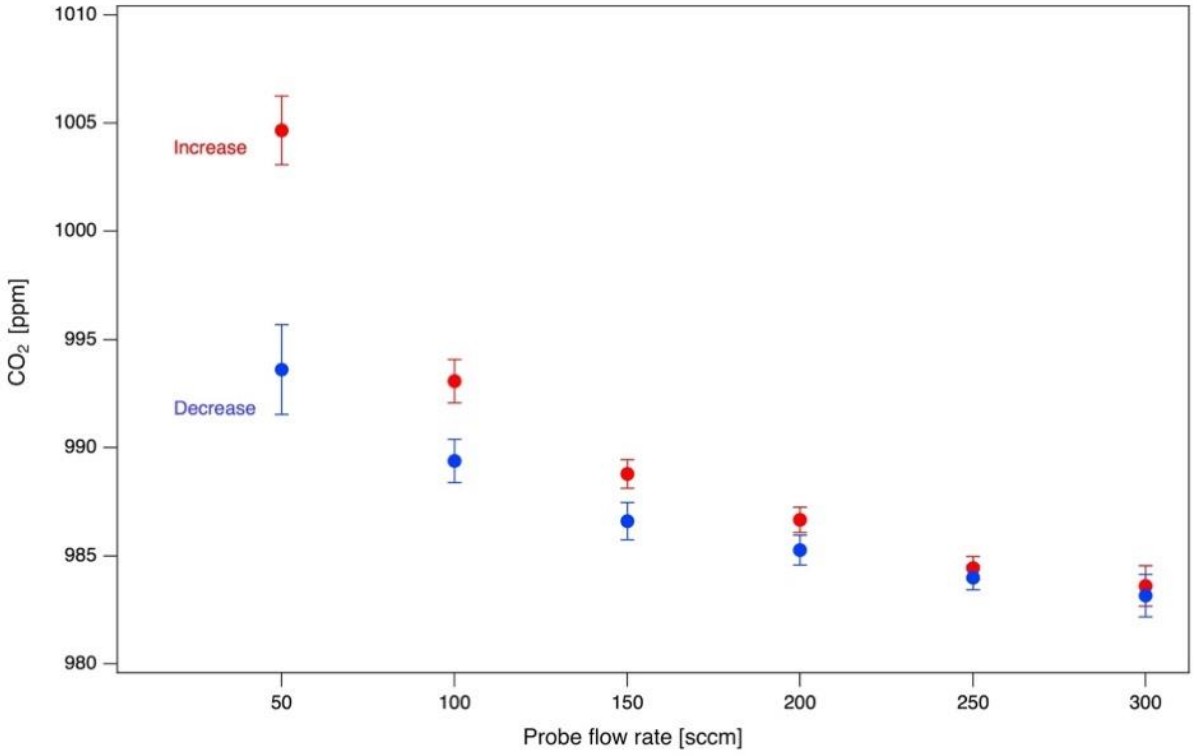

**Figure 4.** Effect of probe flow rate on column gas concentration (System 1 Dual). Points represent concentration of $CO_2$ in the headspace column for one hour after a 15-min probe sampling event at various increasing (forward) and decreasing (reverse) probe sampling flow rates.

### 3.2.2 Impact of probe flow rate and dilution (Experiment 2)

Compared to the controlled soil gas concentrations (Fig. 5), the probe-sampled concentrations were lower. When probe carrier gas is not flowing, the volume inside the probe is fully equilibrated with soil gas. This resulted in the observed initial 'pulse' of high gas concentrations when a probe was first selected and measured. During sampling, probe gas concentrations drop to a steady-state value that represents a balance between probe flow rate and the diffusion rate of soil gas molecules into the probe.

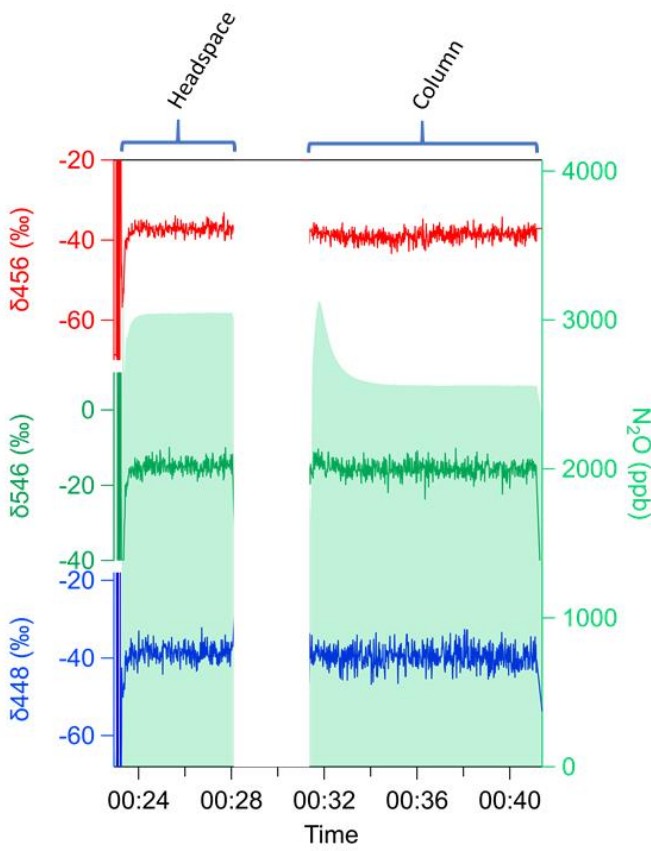

**Figure 5.** Headspace and probe measurements of $N_2O$ using silica in System 2 ($CH_4/N_2O$). Example of initial pulse that equilibrates under flow-through and incomplete diffusion of $N_2O$ concentration (green shade) with undetectable isotopic fractionation of isotopomers □456 (red), □546 (green), □448 (blue).

Gas samples obtained by probes at low probe flow rates were most representative of soil gas, as the slower flow rates allow more complete diffusive equilibration. We evaluated the impact of combinations of different total flow rates (from 50 to 300 sccm at 50 sccm increments) with sample dilution ratios (from 0 to 90% dilution at 15% increments) resulting in probe sampling flow rates between 5 and 300 sccm. These tests were conducted in the silica matrix with controlled soil gas composition (1000 ppm $CO_2$) (Experiment 2). We found that observed soil probe concentrations decreased with increases in probe flow rate (Fig. 6, Fig. 7), with no systematic influence of the dilution ratio. For the probe tested (Table 4), flow rates below 24.5 sccm produced representative samples (within 90% of true concentration). We did not observe any clear drawbacks to sampling $CO_2$ at flow rates <50 sccm (Fig. 7).

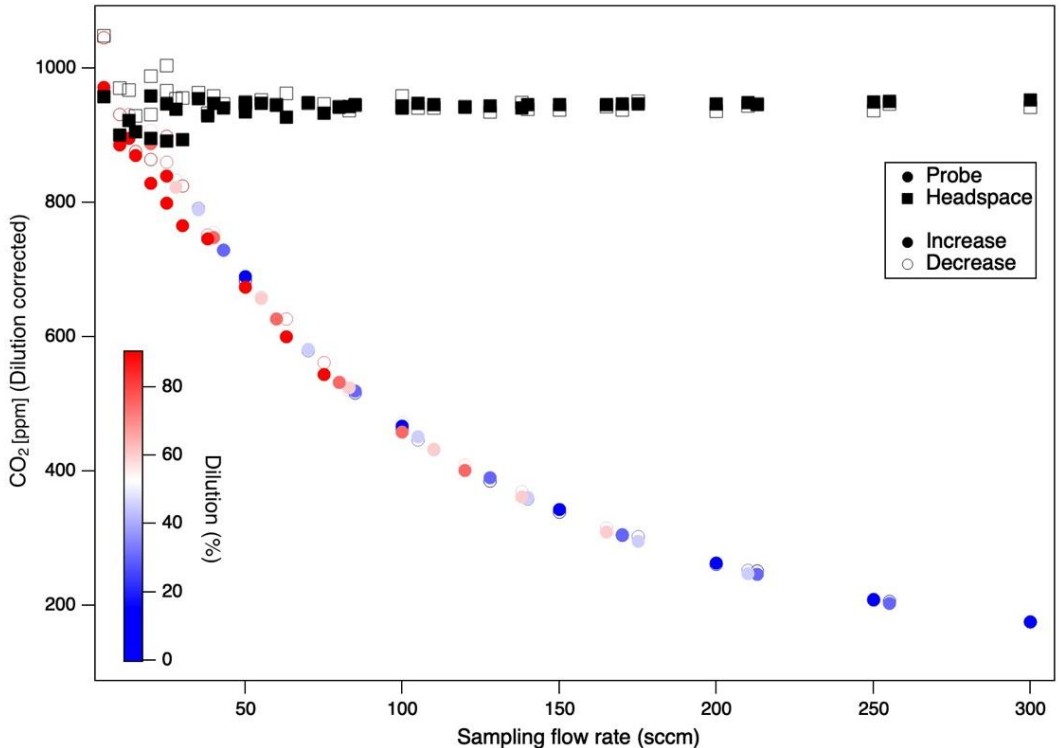

**Figure 6.** Probe and headspace $CO_2$ over a range of probe flow rates and dilution ratios (color). Column soil gas concentrations (headspace) remained steady across the experiment, while gas concentrations sampled by the probe diverged from true values at high probe sampling flow rates. Similar patterns were observed for independent experiments run with the reverse sequence from low to high *vs.* high to low probe flow rates (open vs closed symbols). $CO_2$ concentrations are dilution corrected (System 1 Dual).

Probe flow rates affected gases unequally, and based on their diffusivity. Probe recovery was lower for $CO_2$ with lower diffusivity than CO (molecular diffusion coefficients in air at 20°C: ($CO_2$ 0.14, CO 0.18) (Bzowski et al., 1990; Massman, 1998) (Fig. 7). The fractional recovery of true soil gas concentrations by probe gas sampling (i.e. probe:column headspace ratios) was higher (0.65) for CO than $CO_2$ (0.2) at high flow rates (300 sccm). Additionally, the recovery ratios at specific flow rates were more scattered at a higher flow rate for CO. Regardless of the diffusion coefficient, both $CO_2$ and CO reached equilibrium at low probe flow rates, but CO was well-equilibrated over a 4x wider range (5-100 sccm) than $CO_2$ (5-25 sccm). Moreover, for molecular isotopologues (e.g. $^{12}CO_2$ vs $^{13}CO_2$), at increasing probe flow rates, the sampled $CO_2$ $\square^{13}C$ appears to be lighter than the headspace control by ~ -6 ‰ (Fig. 8) at the highest probe flow rates; showing that with incomplete equilibration, lighter isotopologues with higher diffusivity preferentially diffuse into the probe (here $^{12}CO_2$ vs $^{13}CO_2$). That this fractionation was observed relative to the headspace measurements implies it is derived from the probe, rather than the



rest of the sampling system (tubing, multiport valves, MFCs). These concentration and isotopic fractionation results underscore

the need to ensure that the probe flow rate is sufficiently low to ensure full diffusive exchange between zero air and soil gas

before the gas sample exits the probe.

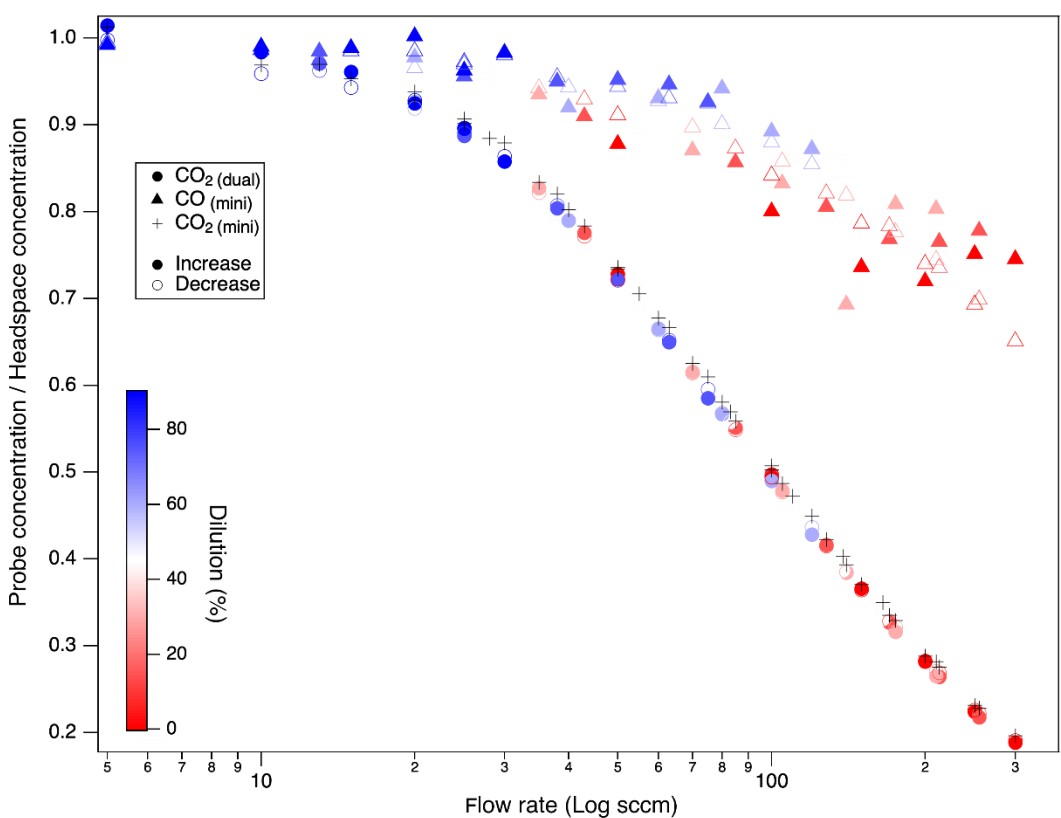

**Figure 7.** Impact of probe sampling flow rate on the fractional recovery of true gas concentrations by probe gas sampling for trace gases with differing diffusivity (CO > $CO_2$) respectively, represented as the fractional recovery (probe:headspace concentration ratio) during a test with a sequential increase in probe flow rate (forward in filled symbols) followed by a test decreasing (reverse in open symbols) the flow rates. Dilution corrected $CO_2$ and CO on System 1 Mini and Dual TILDAS.


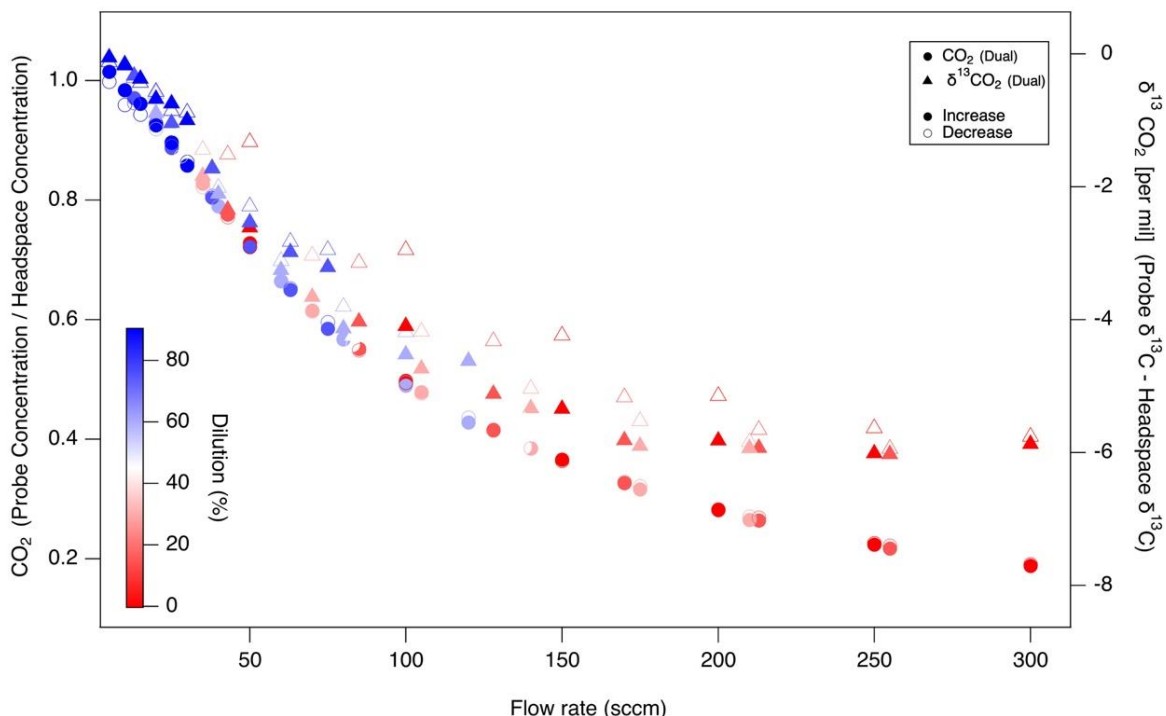

Figure 8. Impact of probe sampling flow rate on the fractional recovery of true $CO_2$ concentrations (left axis, circles) and the offset in true soil $\square^{13}C$ (right axis, triangles) by probe gas sampling. As in Fig. 7, sequential probe flow rate increases (filled symbols) and decreases (open symbols) tests plotted together. Dilution corrected in System 1 Dual.

### 3.2.3 Demonstration with multiple probes (Experiment 3)

We up-scaled the online diffusive probe sampling method in both System 1 and 2 to automatically control multiple probes using lower flow rates (<100 sccm) to measure soil gas concentrations and isotopic ratios (Figure E). To fully constrain probe measurements in the silica matrix (Table 3), each probe was evaluated repeatedly over a full sampling cycle (~25 minutes) to measure headspace-probe-headspace. In both systems, we could scale to sequential measurements of multiple probes with good sample recovery (e.g. minimal concentration loss, isotope fractionation). In particular, probe recovery of $N_2O$ isotopomers was within 3‰ from true headspace values, and equilibration of all trace gas species generally was near or above 85% (Fig. 9). Multiprobe tests showed that the system has a high potential for scalable spatial resolution and scalability.



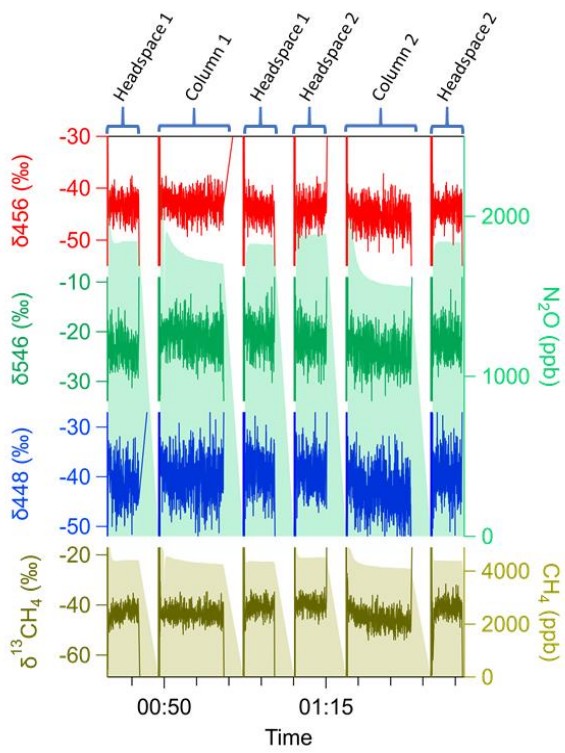

**Figure 9.** Soil probe sampling approach up-scaled to multiple probes (System 2). Multiprobe tests measured headspace-probe-headspace sequentially for (top panels) $N_2O$ (green shade; right side) including isotopic ratios for three $N_2O$ isotopomers $\delta456$ (red), $\delta546$ (green), $\delta448$ (blue) and (bottom panel) $\delta^{13}C$-$CH_4$ (brown; left axis) and $CH_4$ (brown shade; right axis) in the left axis.

We used the multiprobe system to determine whether probes with different properties would exhibit the same flow dependency, and in particular, the effect of characteristic pore size of a sPTFE probe upon concentration recovery. The flow rate dependence of the different probes was determined with $CO_2$ in silica sand (Fig. 10). Our observations of probe flow rate dependency for one pore size (P1) predicted the general behavior of others (P2-P3) across a 5-10 µm pore size range. Unexpectedly, we did not find a clear link between the pore size and the fractional recovery of true soil $CO_2$ concentrations for any given flow rate. For example, we might expect that a pore size of 10 µm would permit greater diffusion and favor probe equilibration; instead, the 8 µm probe produced a more equilibrated sample than either the 5 µm or 10 µm (Fig. 10. Because the pore density of the different probes is unknown, we cannot infer the relationship between the diffusion properties of the probes and the characteristic pore size.

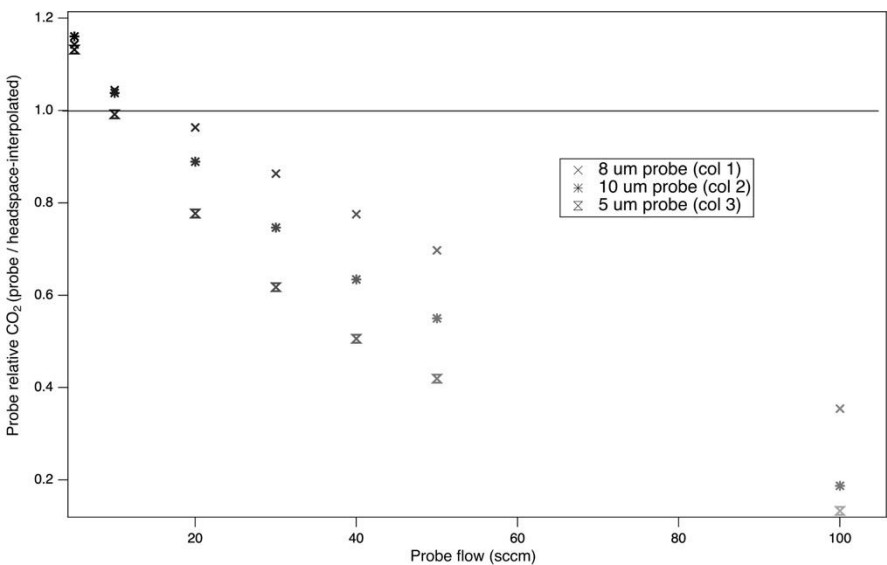

**Figure 10.** Impact of probe pore size on the relationship between probe sampling flow rate and fractional recovery of true soil gas concentrations. Multiprobe test with System 1 Dual. Each Column headspace-probe-headspace were measured sequentially, and headspace values were interpolated to calculate the fractional recovery.

### 3.3 Probe performance in soil

### 3.3.1 Impact of probe flow in soil vs silica (Experiment 3 and 4).

In System 2, at low probe flow rates the concentration measured from the probe was similar to the concentration in the headspace in the silica matrix. Probe flow rates above 25 sccm decreased probe concentration for both the 10 μm and 25 μm pore sizes (Fig. 11). Similar to Fig. 10, the fractional recovery did not increase with pore size, and we did not find that the 25 μm pore size transferred more gas into the carrier flow. In tests at higher probe flow in the silica matrix, the fraction of $CH_4$ recovered in the probe was higher than for $N_2O$. This result was consistent with our results in Fig. 7 and the known molecular diffusion rates of $N_2O$ and $CH_4$ through soil, 0.14 cm$^2$ s$^{-1}$ and 0.19 cm$^2$ s$^{-1}$, respectively (Wang et al., 2014). Thus $CH_4$ diffuses into the probe and replenishes the area around the probe more quickly during sampling than $N_2O$.

In System 2, even in soil where controlled soil gas conditions were lacking (i.e. cannot constrain with headspace measurement), we observed a decline in measured soil gas concentrations with flow rate, similar to the silica matrix experiments (Table 3). The consistency in patterns across systems and soil matrix (controlled silica vs uncontrolled soil) provided additional support for the trends observed in the above experiments. The experiments that follow did not attempt to control soil gas concentrations in real soil and focused on probe gas and not headspace concentrations. In the following tests, we manipulated key drivers of soil function (moisture and redox) to elicit responses in soil microbial processes and soil gas concentrations to discover the in situ soil gas dynamics newly observable with our novel soil gas probe sampling system.



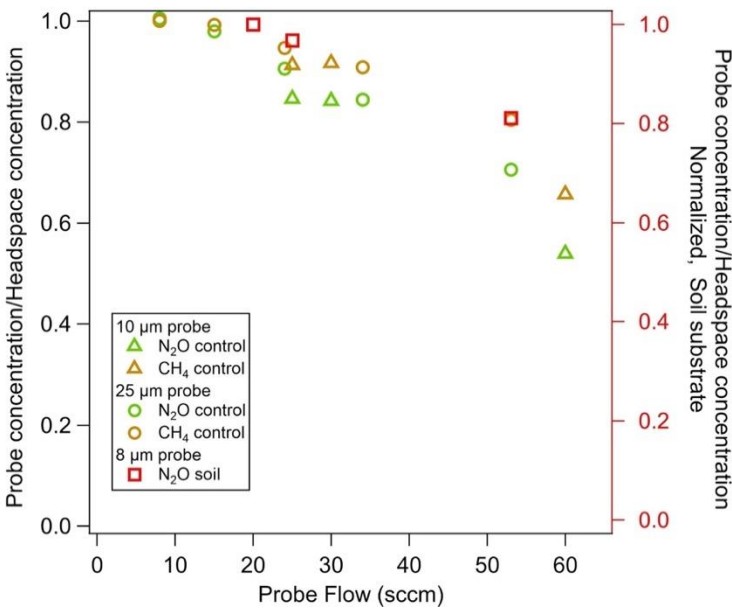

**Figure 11.** Impact of probe sampling flow rate, pore size, trace gas species, and soil matrix on the fractional recovery of true soil gas concentrations with probes. Fractional recovery of $N_2O$ (green) and $CH_4$ (yellow) in a silica matrix with flowing control gas and probe pore size of 10 µm (triangle) and 25 µm (circles). The recovery of $N_2O$ gas in soil at field moisture (red squares), normalized to high recovery, measured with probe pore size 8 µm. All measurements using System 2.

**3.3.2 Soil dry-wet cycle (Experiment 5)**

Soil wetting induced a strong pulse in subsurface $N_2O$ concentrations, isotopic signatures, and site preference that was captured in detail with the $N_2O$ and $CH_4$ TILDAS and real time in situ soil gas probe sampling. We found that the isotopic ratios of all three $N_2O$ isotopomers ($\delta448$, $\delta546$, $\delta456$), site preference, and $N_2O$ concentration responded to the wetting over the subsequent 36-hour period. $N_2O$ rose from approximately 3 ppm to over 40 ppm, with a corresponding and slightly delayed response in isotopic signatures (Fig. 12). The dramatic increase in $N_2O$ required additional dilution at concentrations above the expected range of the TILDAS (>20 ppm). The response of the two $^{15}N$-N2O isotopomers diverged enough to drive a shift in the site preference (SP) upward by approximately 4‰ to 6‰ before falling back down toward 2‰. After the peak, the decline in concentration and isotopic signatures was not explained by soil moisture, which was a relatively steady 25-30% volumetric water content (VWC) throughout the period. When mapped into a 3-dimensional isotope space (Fig. 12b) that is based upon previous observations of SP, $^{15}N_{bulk}$, and $^{18}O$ for a variety of different processes (Toyoda et al., 2017; Wei et al., 2019), the observed isotopic signature falls between chemodenitrification and bacterial denitrification. While the $^{15}N_{bulk}$, and $^{18}O$ signals are dependent upon the substrate $^{15}N$ and $^{18}O$ compositions, the shift over the course of the rewetting measurement indicates a period of more denitrification (at higher SP), then decreasing back to bacterial denitrification. Importantly, the





observed range of SP values is well below the expected range for bacterial and archaeal nitrification (AOB, AOA), which are
>20 (off scale in Fig.12b).

In contrast to the dynamic response in $N_2O$, soil $CH_4$ concentrations remained low, leading to low signal-to-noise

ratios in the detected $^{13}C$-$CH_4$ isotopologue, and did not respond to wetting (data not shown). The dilution rate of the sample
was increased by 1.9x at hour 18, resulting in a 1.9x reduction in $N_2O$ concentration measured by the TILDAS (accounted for
in Fig. 12). Despite the large change in concentration, the isotopic signatures barely changed, even after readjusting the dilution
rate at hour 42, indicating that their concentration dependence had been well accounted for.

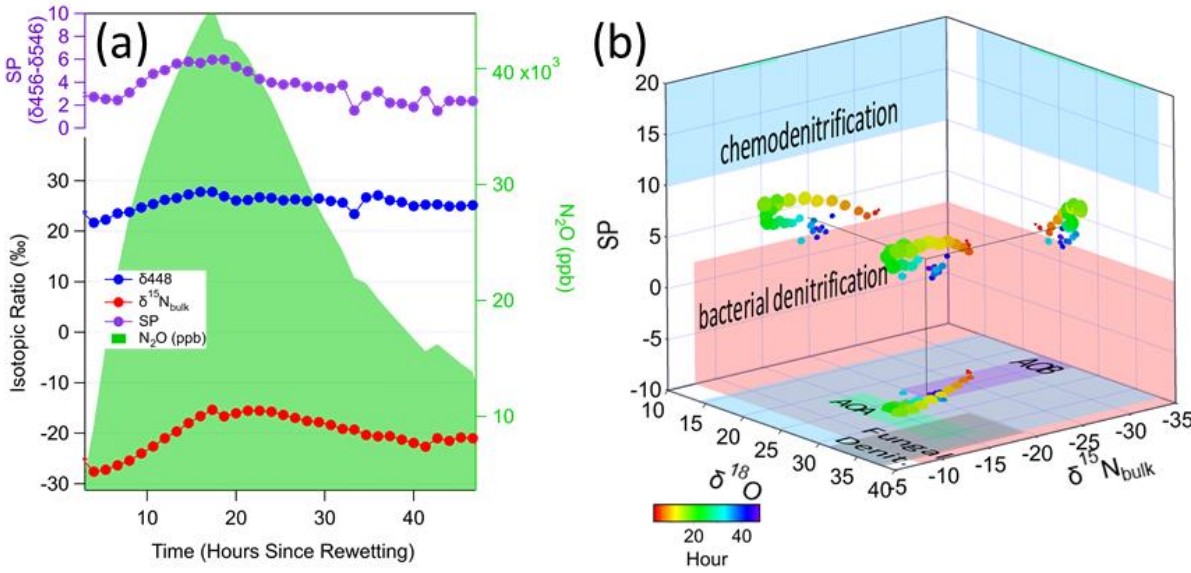

**Figure 12. (a)** Soil wetting induced a pulsed response in soil $N_2O$ (shaded green) and its isotopic signals including δ448 (blue),
δ546 (green), δ456 (red), and site preference (purple). A soil column without a lid was wetted with the equivalent of 5.1 cm of
rainfall. At 18 hours after wetting the dilution was changed from 2:1 to 3.8:1, and at 41 hours it was changed to 2.1:1, which
is accounted for in the concentrations reported here. **(b)** Estimated map of $N_2O$ isotopic signatures of bulk $δ^{15}N$ (x-axis), $δ^{18}O$
(y-axis), and site preference (z-axis), circles represent probe measurements of the changes in the isotopic signatures with time
(hours) indicating shifts into region of different microbial activity (colored rectangles) (Table S3). On the x-axis AOA (green
rectangle) and AOB (purple rectangle) refer to nitrification from ammonia oxidizing archaea and ammonia oxidizing bacteria,
respectively. Grey rectangle indicates fungal denitrification.
**3.3.3 Soil responses to an anaerobic to aerobic shift (Experiment 6)**

Shifting the soil redox environment from anaerobic to aerobic conditions induced a cascade of subsurface gas pulses

in $CO_2$, $N_2O$, and VOCs that we measured by integrating TILDAS and Vocus analyzers with the real time in situ soil gas probe



sampling. Before this experiment, the soil column was forced into anaerobic conditions by advectively flushing with $N_2$
through the control gas ports for 3.5 hours; subsequently, conditions were driven aerobic by flushing the system with UZA for
a short time at time zero (Fig. 13). Conversion to aerobic conditions drove a pulse in $N_2O$ concentrations that was slow and
considerably weaker (reaching 1.6 ppm after 72 hours) than the wetting response (Experiment 5). The onset of aerobic
conditions brought a strong $CO_2$ increase from 0.1 to 0.4%, suggesting an increase in microbial respiration. Along with $CO_2$
and $N_2O$, we measured a cascade of responses in masses corresponding to different VOCs. As respiration and nitrogen
processing increase, the larger VOCs exhibit either immediate ($C_9H_{18}O$, $C_{11}H_{20}O$, e.g. nonanal, methylborneol) or delayed loss
($C_{10}H_{16}$ (monoterpenes), $C_{12}H_{22}O$, e.g. geosmin) in the soil. In contrast, after a few hour delay, the sulfur-containing
compounds methanethiol ($CH_4S$) and dimethyl sulfide ($CH_6SH$) exhibited a surge in production. The approach captured
different temporal responses to a shift in soil redox across a suite of soil gases that reflect different microbial processes and
their sensitivity to environmental forcing.

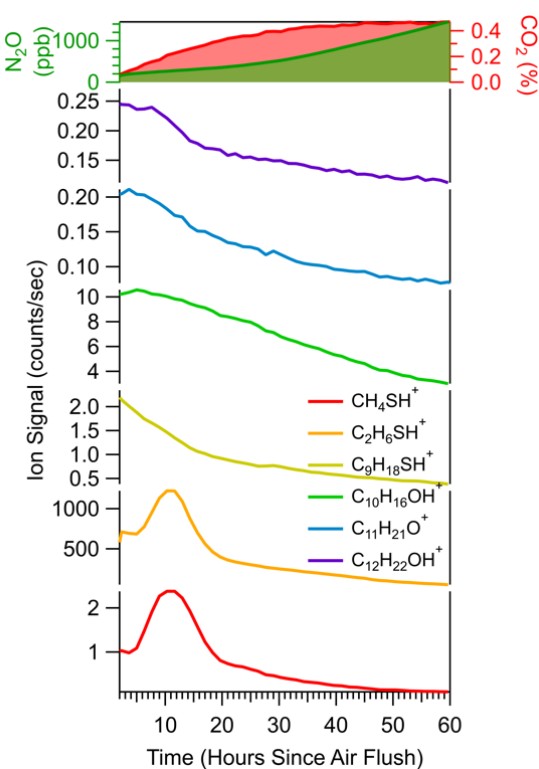

**Figure 13.** A sudden change from anaerobic to aerobic soil conditions, induced by flushing with UZA, drove dynamic
responses in $N_2O$, $CO_2$, and a variety of VOCs captured using the diffusion-based soil probe integrated with the TILDAS and
Vocus analyzers. System 2 Experiment 6 with a B2 TRF soil sample.



## 4. Discussion

We developed a new soil gas sampling system that integrated diffusive sPTFE soil probes with a set of online and high resolution trace gas analyzers. The versatile system detected controlled and forced temporal changes in soil concentration and isotopic signatures of $N_2O$ and $CH_4$ and VOCs.

### 4.1 Optimizing soil gas sampling

*Optimizing soil gas sampling.* Our results show that gas recovery depends on probe flow rate and the trace species, while the effect of dilution of the probe sample outflow on recovery is minimal. Probe flow rate determines the time available for carrier UZA to equilibrate with soil gas across the diffusive membrane as it flows through the probe: lower probe sampling flow rates allow more time to equilibrate than do high flow rates (Gut et al., 1998; Parent et al., 2013). By running tests in reverse order, we showed that the results were not dependent upon carry-over or memory effects. Correspondingly, we observed that the fractional recovery of true soil gas concentrations declined exponentially with increased probe flow rates across all systems (Fig. 8 and Fig. 11), analytes (Fig. 7), and probe characteristics tested. The maximum probe flow rates that delivered well-equilibrated samples (>90% equilibrated) ranged from ~25 to 100 sccm, depending on the system and, in particular, the molecule measured. Indeed, in both silica and soil matrix, gas recovery was better for molecules with relatively higher molecular diffusivity (i.e. CO, $CH_4$, $^{12}C$-$CO_2$) than paired analysis of those with lower diffusivity (i.e. $CO_2$, $N_2O$, $^{13}C$-$CO_2$) (Wang et al., 2014). Molecules with higher diffusivity move across the membrane and also replenish the area around the probe during sampling more quickly than those with lower diffusivity. As a result, the upper range of probe flow rates that produce representative gas samples will be higher for analytes with higher diffusivity, and more restricted for slow diffusing molecules. While isotopic fractionation was observed in some ($CO_2$; Fig. 8), but not all ($N_2O$; Fig. 9) tests, incomplete equilibration affected recovery of bulk concentration more strongly than isotopic signature, suggesting that optimized probe sampling can produce isotopically representative samples with minimal fractionation.

### 4.2 Factors yielding a representative sample

One of the challenges in soil trace gas measurements is transferring a representative sample (Parent et al., 2013) from probes to fill the relatively large sample cell volumes of online analyzers (e.g. 10s to 100s mL at reduced pressure). To address this issue, we reduced the effective volume of the TILDAS sample cell by designing a more compact cell with a volume-filling insert (Section 3.1). We also integrated online dilution into the sample transfer system after the probe, which increased the sample volume delivered to the sample cell without increasing probe flow rates. Dilution also helped reduce soil gas concentrations to within the range of sensitive trace gas analyzers and avoid condensation effects (none observed). Together, these modifications improved the transfer of representative soil gas samples to the cell, increased the turnover time of the cell leading to a faster time response, and supported lower probe flow rates facilitating probe equilibration (Jochheim et al., 2018).



Beyond flow-through sampling, these modifications may be particularly important in future approaches that transfer
equilibrated soil gas 'plugs' to an online analyzer for trapped-sample analysis. In addition, reducing sample demand also
reduces the disruption of the soil probe measurement on the soil environment. The diffusive soil probes allow sample gas to
diffuse into the probe from the soil environment, but also allow the UZA carrier gas to diffuse out of the probe into the soil.
Under controlled soil conditions (silica and advective flow), probe sampling caused a $< 2\%$ decrease in soil $CO_2$ concentrations,
with the impact decreasing at the low probe flow rates supported by our volume-reducing modifications. In real soil, the impact
of carrier diffusion out of the probe could be larger where local gas concentrations are not replenished by advection but depend
on local production, consumption, and diffusion. In addition to reducing sample volume, lowering the sampling frequency
(return rate) may be especially important in real soil (Jochheim et al., 2018) helping to reduce the impact of the perturbation
on the soil environment.

**4.3 Transferability to multiple analyzers**

The continuous online soil gas sampling approach is highly transferable across trace gases and instrument systems.
Here, we successfully measured soil trace gases using two systems. Modifications to reduce sample volume requirements (i.e.
online dilution, precise flow control, instrument modifications) are transferable to other analyzers beyond the $N_2O/CH_4$
isotopologues TILDAS. Although other laser absorption spectroscopy instruments like CRDS have been used to measure
concentration and isotopic composition for trace gases like $CO_2$ (Voglar et al., 2019), TILDAS can measure several species at
high sensitivity/spectral resolution with one instrument (McManus et al., 2015), are field deployable (McCalley et al., 2014;
Roscioli et al., 2015; Saleska et al., 2006), and readily interface with the valving and flow control system designed here. Not
only is the approach transferable across instruments, but we demonstrated that more than one instrument can be integrated for
simultaneous soil probe sampling, e.g. Vocus PTR-TOF-MS for VOCs with the $N_2O/CH_4$ TILDAS in parallel (System 2), and
two TILDAS in series (System 1). This versatility can be extended to allow analysis of a suite of soil gases using existing
TILDAS technology to study, for example, soil microbial N cycling (e.g. $N_2O$, NO, $NO_2$, $NH_3$, $HNO_3$, HONO, $NH_2OH$),
microbial trace gas scavenging (e.g. CO, OCS, $CH_4$, $O_2$), and other atmospherically-relevant species (e.g. $H_2O_2$, HONO, $N_2H_4$,
HCHO, HCOOH, $CH_3OH$). These compounds represent metabolites for microbial communities, and intermediates of
metabolic pathways of carbon and nitrogen cycling. Therefore, coupling these instruments with soil probes will enable access
to previously unexplored biological information that reflects metabolic and signaling processes in the soil subsurface.

**4.4 Capturing soil gas response to wetting**

The optimized soil gas sampling system integrated with the novel $N_2O/CH_4$ TILDAS captured real time responses of
subsurface $N_2O$ isotopes to changes in soil moisture in soil column experiments. Soil wetting is a powerful and well-studied
driver of biogeochemical change in soils known to result in a rapid release of soil gases such as $N_2O$ (Birch effect) (Birch,
1958; Leitner et al., 2017). The soil probes, positioned at 20 cm below the soil surface, captured a significant increase in



subsurface N$_2$O concentration almost immediately after water was added to the column, and a slow change in isotopic signature
that suggests a more gradual change in the subsurface processing producing N$_2$O (Leitner et al., 2017; Van Haren et al., 2005).
Our novel subsurface $^{15}$N site preference measurements had SP signatures for N$_2$O production pathways between characteristic
signatures suggested for bacterial denitrification and chemodenitrification pathways (Sutka et al., 2006; Toyoda et al., 2017).
The rapid increase and slower decrease of N$_2$O suggest the initial pulse of microbial activity as a response to environmental
changes can vary on long (days) and short timescales (minutes/hours) at this depth, both of which are recovered using this
sampling system.

**4.5 Capturing soil gas response to redox**

The system displayed the potential to capture hotspots or hot moments of trace gas production, providing temporal
information of key metabolic processes in carbon and nitrogen cycling. The introduction of UZA to change the redox status
increased the abundance of microaerobic sites in soil stimulating heterotrophic respiration (CO$_2$ emissions). VOC production
also responded to the changes in oxic and anoxic sites. VOCs are emitted as a byproduct of either respiration or fermentation
processes as part of microbial metabolism in soil (Peñuelas et al., 2014). Small molecules and larger volatile organic
compounds contribute to soil nutrient cycling, and therefore serve as valuable markers of different and highly specific
microbial activity (Schulz-Bohm et al., 2015). For example geosmin and methylisoborneol are produced by actinomycetales
(Citron et al., 2012; Peñuelas et al., 2014) under anaerobic conditions, while sulfurous VOCs are produced in micro-anaerobic
sites in soil.

**4.6 Implications for sPTFE as a field-based soil probe**

sPTFE is known to exhibit hydrophobicity, chemical inertness, microfiltration properties, and uniform pore
distribution (Dhanumalayan and Joshi, 2018). All sPTFE probes recovered representative soil gases, but without differences
based on pore size. Sample recovery did not correlate with the characteristic pore size of the probes, and all sizes quantitatively
recovered >90% of the analyte concentration at optimized flow rates. It is unknown whether the machining process modified
the pore density of the surface of the probe; nevertheless, it did not damage the integrity or the resistance of the material, which
preserved hydrophobicity and structure throughout the experiments. In >4 months of operation in laboratory soil, the
performance of the sPTFE probes did not change with time or environmental conditions. In contrast, Panikov et al., 2007 found
that the methane conversion factor for calibration using silicone membranes differed between a dry and wet membrane.
(Rothfuss and Conrad, 1994) found memory effect issues when sampling high concentrations of CH$_4$ with silicone and epoxy
as soil-gas exchange barriers. Soil probes made of polypropylene (PP) membranes have been widely used to measure CO$_2$
(Gut et al., 1998; Jochheim et al., 2018),  and polyethylene (PE) for water isotopes in soils (Volkmann and Weiler 2014;
Volkmann et al. 2018) and in tree xylem (Volkmann et al., 2016a). However, in our past experience (T. H. M. Volkmann,
personal communication) PP and PE probes have shown decreased wall integrity during field deployment and long term use





(i.e. dents and cracks) causing gas and water leaks, compromising hydrophobicity in saturated media. Our 15 cm probes are
more rigid and smaller than previous probes that were typically 100 to 150 cm in length (Flechard et al., 2007; Gut et al., 1998;
Parent et al., 2013; Rothfuss et al., 2013), and are easily installed via a small borehole. The sPTFE soil probes described here
therefore have potential to be less disruptive to the soil ecosystem and more robust to soil structure and environmental changes
for long-term measurements in the field.

**4.7 Considerations for field deployment of the system**

We show that diffusive soil probes can measure soil gases with cm-level spatial resolution for a time-dependent
picture of the soil gas dynamics. This contrasts with other methods, e.g. manual sampling with syringe (Kammann et al., 2001)
and cartridges (Wester-Larsen et al., 2020), that disturb the true soil gas concentration during sampling and also risk sample
integrity during transfer for offline laboratory analysis (Volkmann and Weiler, 2014). Manual sampling also increases the
potential measurement error, and is time consuming and labor intensive, particularly when attempting to sample at a high time
resolution and/or to cover a large spatial area (Wester-Larsen et al., 2020). Our integrated sample system can achieve
unattended, automated sequential and long-term field soil gas sampling that is less time consuming and less laborious.
In field implementation of our system, there are tradeoffs between sampling frequency and disruption that should be
fully considered. As noted above, diffusive soil sampling can alter soil gas by dilution, and sample transfer parameters should
be optimized to obtain representative samples with minimal disruption. This may be especially important for distant sampling
points that require longer tubing that may release more zero air into the soil during sample transfer to the analyzer. The different
modules of the sampling system (Fig. 2) are flexible and can be adjusted to accommodate multiple probes, different
measurement specifications, and soil and environmental factors in the field.

**5. Conclusion**

Versatile trace gas sampling systems integrating soil probes and high resolution trace gas analyzers bridge an existing
knowledge gap using in situ spatial (centimeter scale) and temporal (minutes) measurement of concentrations and isotopic
signatures of soil trace gases. We demonstrated the feasibility and versatility of an automated multi-probe analysis system for
soil gas measurements of isotopic ratios of nitrous oxide ($\delta^{18}O$, $\delta^{15}N$, and the $^{15}N$ site-preference of $N_2O$) and methane ($\delta^{13}C$),
and VOCs, all important gas-phase indicators of biological activity. We present an experimental system to evaluate and
optimize probe sampling under controlled conditions and demonstrate capabilities to resolve dynamic changes in real soil. The
experimental approach captures snapshots of gas emissions as a result of changes in environmental drivers such as soil moisture
and redox conditions, and observed hot moments showing the dynamics of microbial metabolism and communities. These
tests demonstrate the potential of this approach to reveal interconnections between the soil microbiome and its local
environment on timescales relevant to real-world variability.



In order to keep expanding the usefulness of these probes, future work will focus on further characterization of probe
material, pore size, and dimensions, with the goal of reducing size (and therefore sampling footprint) in order to access
processes on smaller spatio-temporal scales. We will also explore probe assembly and installation approaches that minimize
disturbance to the subsurface, and allow for rapid installation into soil.
The outlook is bright for integrating soil gas measurements with other data and models to unlock new understanding
of soil microbial processes. Direct sampling of soil for subsequent laboratory incubations and analysis using multi-omic
approaches is a sensitive and precise approach for identifying subsurface microbial populations and their potential metabolic
function. Although both widely used approaches produce reliable and robust results, they are labor intensive and destructive,
and incompatible with generating a well resolved spatial- and time-dependent understanding of microbial activity in natural
ecosystems. Similarly, current soil gas sampling methodologies face challenges to address the gap between time-space
sampling (e.g. frequency and intensity), low bias in downstream analysis, and proper reference materials. Isotopic signatures
of trace soil gases, in conjunction with genomic and metabolomics approaches can elucidate real time biomarkers of microbial
metabolisms in soil, leading to a better understanding of soil heterogeneity as a modulator of soil-microbe interactions and
their responses to environmental factors and nutrient cycling. These efforts will help scale up soil trace gases monitoring and
quantification of biogeochemical processes to improve modeling, soil management decisions, and soil health with high spatial
and temporal resolution.
**Data availability.** Igor software was used under license. Igor scripts were used for data processing and analysis including
Aerodyne Research Inc. proprietary scripts for parsing and averaging data and cannot be in a public repository. Other portions
of Igor code used for plotting are available upon request. Raw measurements files (e.g., TILDAS and vocus spectra) will be
available upon request. Processed data can be found at DOI: 10.25422/azu.data.13383014
**Supplement.** Additional supporting information available online at:
**Author contribution.** All authors made substantial contributions to the research. T.H.M.V, L.K.M., J.R.R., J.H.S.
conceptualized the idea and acquired funding. All authors participated in part or all of developing prototypes, building
experimental systems, and conducting experiments. J.G.L, L.K.M., J.R.R., J.H.S. contributed to the analyses and interpretation
of data; J.G.L. and L.K.M. prepared the draft, all authors discussed the results and contributed to the final manuscript.
**Acknowledgments.** This material is based upon work supported by the U.S. Department of Energy, Office of Science, Office
of The Small Business Innovation Research (SBIR) grant, Office of Science, under Award Number(s) DE-SC0018459. THMV
was supported by Biosphere 2 through the office of the Senior Vice President for Research Innovation and Impact at the
University of Arizona. We thank Doug White and White Industries, Inc. for machining the probes. The authors gratefully





acknowledge financial support from the Philecology Foundation for Biosphere 2 and the Landscape Evolutionary Observatory. Prof. Shuhei Ono at the Massachusetts Institute of Technology has shared with us calibrated reference gases for this study.

**Conflicts of Interest.** Aerodyne Research Inc manufactures the TILDAS instrumentation and commercializes the Vocus PTR-TOF for applications in geosciences. Probes, sampling systems and associated software are in development.

Disclaimer: *"This report was prepared as an account of work sponsored by an agency of the United States Government. Neither the United States Government nor any agency thereof, nor any of their employees, makes any warranty, express or implied, or assumes any legal liability or responsibility for the accuracy, completeness, or usefulness of any information, apparatus, product, or process disclosed, or represents that its use would not infringe privately owned rights. Reference herein to any specific commercial product, process, or service by trade name, trademark, manufacturer, or otherwise does not necessarily constitute or imply its endorsement, recommendation, or favoring by the United States Government or any agency thereof. The views and opinions of authors expressed herein do not necessarily state or reflect those of the United States Government or any agency thereof."*

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
