# Peer review of "Versatile soil gas concentration and isotope monitoring: optimization 1"

_Biogeosciences, 2020_

## Referee Comment (RC1) · Anonymous Referee #1 · 13 Feb 2021

The soil gas concentration and isotope composition monitoring device that is presented in this manuscript is really promising. The system is described with a lot of details, which is important. The device has been carefully checked and the first experiments reported show well the potential of this type of approach in the field of soil science and biogeochemistry. I have only few comments, requests or questions below. I have to confess that I am biologist, not physicist.

The importance of storage of gas in the air-filled soil porosity on the link between soil concentration and net surface fluxes, and the dynamics of the air-filled soil porosity, at

short term (change in water content), long term (e.g. compaction), and also its spatial variation (e.g. local variation in bulk density), are not enough clearly stated and can be highlighted in the first paragraph of the introduction.

Tracing the fate of labelled gas in soil profile is also another promising approach that can be achieved by the development of such integrated monitoring tools. This can be mentioned in the second paragraph of the introduction.

The carrier gas (Ultra Zero Air) contains $O_2$ which diffuse to the soil. Is there a risk that microbial processes under study are altered by this change in $O_2$ concentration in the soil? Would it be better to use pure $N_2$ instead? Would it be possible to use a close loop system rather than an open one? Can you discuss your choice for an open-one?

The control gas that were used for the test using silica-filled column looks a bit low compared to expected soil concentration. Would the results of the test be different with $CO_2$ or $CH_4$ concentration of 1% or more?

Can you elaborate a bit more and give a definition of "backgrounds" (L257-261)

You confirm that lower probe sampling flow rates allow more time to equilibrate than do high flow rates. Would it be possible to use the initial 'pulse' of high gas concentrations at the beginning of the measurement with high flow rates until the steady state value is reached to recompute the initial concentration in the soil air (peak integration)?

Details

Section 3 title is "results and discussion" but section 4 is "discussion". Should section 3 be "results" only? Sometime, delta is not well converted in the pdf and become a square instead

---

## Referee Comment (RC2) · Anonymous Referee #2 · 13 Feb 2021

I would like to congratulate the authors on what I found to be a very interesting and informative paper on a very important topic. The prospect of deciphering the source of gas emissions based on isotope ratios, in-situ and with minimal soil disturbance is very exciting, and I look forward to seeing where this research goes next. I have no major comments, and my minor comments are mostly related to making this more accessible to people less-well versed in these kinds of analyses (like me!).

Minor comments:

Is there still control air coming through the bottom of the column when the probe is incubated in soil, or only for flushing the column for rapid redox state shifts?

Can the authors add in a sentence about the response/equilibrium time (if it can be deduced from the flow rates) for the gasses in the system in the setups shown, and how does that compare to other published probe setups? This would be especially important for highly temporally dynamic and depth-stratified systems.

Why did the authors choose to sample destructively in the soil setup rather than recirculating the air through the column after going through the TILDAS? It doesn't seem like TILDAS is a destructive method so a closed system should be possible (and possibly more desirable) for the soil experiment and more amenable to translating the setup to controlled in-situ studies. I would assume that an open system with fresh ultra zero air would just generate a concentration gradient and accelerate influx, such that the relative difference between actual and perceived gas concentration would be greater at high flow rates compared to low flow rates after accounting for dilution. But that there is also a counteracting equilibration time effect at fast flow rates (which is what is shown in figure 4 and 6, without necessarily parsing out the magnitude of concentration gradient and time to equilibrium effects).

Maybe something to speculate on whether it would be possible to make the detector volumes even smaller so that the probe volumes could be smaller and perhaps depth-resolved, or whether the whole system would need to be re-developed. This would be a really useful for any system where the soil is strongly depth resolved in terms of chemistry and/or temperature (ex thawing tundra or forest floor)

Typographical comments:

Soil texture for S1: if the clay+silt = 66%, then the sand cannot be > 34%. L600 with rather than without?

---

## Referee Comment (RC3) · Anonymous Referee #3 · 18 Mar 2021

1. Does the paper address relevant scientific questions within the scope of BG?

   Without any doubt reliable protocols for diffusive soil-gas sampling combined with multi-component concentration and isotopic analysis are issues in soil soil ecological research. Indeed, progress in field research about soil-gas exchange requires reduced impact diffusive sampling with good temporal and spatial resolution. Another point are costs, that allow installation with sufficient spatial repetition. Therefore the topic of the paper adress clearly aims and scope of BG. My problem with the paper is, that in the study itself the critical issues are not treated

   critically and in comparison to existing solutions. The paper lacks a well defined scientific question. It seems to me more than an advertisement for a specific technical solution.

2. Does the paper present novel concepts, ideas, tools, or data?

   Neither diffusive gas sampling, nor the use of hydrophobic porous tubes are novel concepts. The authors use an obviously sophisticated assembly of gas analyzers covering a wide range of isotopic and concentration responses. However, I miss a comparative discussion about the novelty and the advantages of the used equipment. An interesting concept is presented in figure 12-b, where the isotopic signatures of N2O are plotted in a 3-D coordinate system and assigned to specific processes. However, this process analysis is not further discussed and the background of the interpretations are not reported with exception of a literature overview in the supplementary material.

3. Are substantial conclusions reached?

   No.

4. Are the scientific methods and assumptions valid and clearly outlined?

   The description of the sampling and analysis system is difficult to read and full of technical details that do, however, not focus on critical issues. I clearly miss a list of the requirements that are expected from "a versatile soil-gas concentration and isotope monitoring". Insofar the reader does not really learn, what the authors expect from their equipment with regard to quantification limits, system disturbance, usability in the field, or costs.

5. Are the results sufficient to support the interpretations and conclusions?

   I miss specific scientific interpretations and conclusions with regard to a reference. The extensive report of results is somewhat anecdotal.

6. Is the description of experiments and calculations sufficiently complete and precise to allow their reproduction by fellow scientists (traceability of results)?

   The technical descriptions are very detailed but not in such way, that it could be used to build up a system for measurement in soil columns or even in the field. One of several issues not adressed are power requirements and external conditions to work with the assembly of gas analyzers under field conditions. Neither I found suggestions for low-impact installation of the novel gas samplers or dead volume of the system.. So I have to answer with no.

7. Do the authors give proper credit to related work and clearly indicate their own new/original contribution? The paper is more a technical description or even an advertisement of a specific setup but not mainly a scientific paper. It is not fully clear what is new (the use of PTFE instead of PE porous tubing?) or the use of an assembly of different laser-gas analyzers? Several literature references seem to more be an enumeration than a than a critical review.

8. Does the title clearly reflect the contents of the paper?

   In principle yes.

9. Does the abstract provide a concise and complete summary?

   Yes, including the weaknesses of the scientific substance.

10. Is the overall presentation well structured and clear?

    Too extensive, too complex, lacking structure and conciseness.

11. Is the language fluent and precise?

    Yes.

12. Are mathematical formulae, symbols, abbreviations, and units correctly defined and used?

    Yes.

13. Should any parts of the paper (text, formulae, figures, tables) be clarified, reduced, combined, or eliminated? For publication the whole paper should be focused, shortened and reorganized including the formulation of a clear scientific question that can be discussed based on hypotheses. I would suggest to split up the paper into a physically based sampling optimization part and another, multi component (isotopic) analysis part and possibly the process study of the process interpretation by isotopic signatures.

14. Are the number and quality of references appropriate?

    Yes.

15. Is the amount and quality of supplementary material appropriate?

    Some parts of the supplementary material should be used in the main text. I miss in both, the paper and the supplementary material a listing of quantification limits of the equipment.

---

## Author Comment (AC1) · 1 Apr 2021

Dear Dr. Nicolas Brüggemann and Reviewer #2,

We appreciate the positive comments and the reviewer's #2 consideration of that innovation in soil probes and soil trace gases instrumentation is very important. Following, we will address each of the reviewers comments and suggestions, those that required edits to the manuscripts will be addressed and will be noted in each response if pertinent for the editors' evaluation.

RC2: Is there still control air coming through the bottom of the column when the probe is incubated in soil, or only for flushing the column for rapid redox state shifts?

Answer: The reviewer brings up a good point for us to clarify in the text and Table 4. In tests with silica (Table 3), we continually fluxed the matrix with control gas to provide our 'gold standard' for comparison to soil probe measurements. In soil experiments (Table 4) the control air was added to quickly change redox conditions and stopped prior to measurements. We have clarified these experimental details by adjusting columns in Table 4 to more clearly list the control gas and its purpose.

RC2: Can the authors add in a sentence about the response/equilibrium time (if it can be deduced from the flow rates) for the gasses in the system in the setups shown, and how does that compare to other published probe setups? This would be especially important for highly temporally dynamic and depth stratified systems.

Answer: In the open flow-through probe sampling approach described here, soil probes have time to equilibrate with soil before being sampled, so that when selected, the most fully equilibrated sample is initially observed (e.g., Fig. 5 at time 00:31). As the flow-through sampling progresses, gas concentrations sampled via the probe come to steady state values proportional to soil gas concentration and inversely proportional to flow rate (Fig. 6). In steady state, sampling duration does not increase equilibration and there is no equilibration time we can report.

In contrast, closed-path recirculating flow-through sampling can progressively come closer to equilibration with every pass. For example, DeSutter, Sauer, and Parkin 2006 (doi:10.1016/j.soilbio.2006.04.022) provide a nice evaluation of extruded PTFE vs Polyethylene (PE) tubing in a closed pass setup with a $CO_2$ sensor and report the time of 95% equilibrium (teq) of $CO_2$.

Here, we can calculate the residence time of carrier gas in the soil probe by considering the internal volume of the probes (V=2.6-4.6 mL) and the range of flow rates evaluated (F=5–300 sccm). This indicates that the residence time (V/F) could range

from <1 sec for high flow rates to 55 sec for the lowest flow rates and larger volume (5 sccm in probes P5, P8, P10). We agree with the reviewer that this is helpful information to provide because it helps explain the dependence on probe flow rate (Fig. 6), and we can add the residence times of gas in the probe (lines TBD). We note that even the lowest residence times are actually very fast, and would be suitable for highly temporally dynamic and depth-stratified systems.

RC2: Why did the authors choose to sample destructively in the soil setup rather than recirculating the air through the column after going through the TILDAS? It doesn't seem like TILDAS is a destructive method so a closed system should be possible (and possi-bly more desirable) for the soil experiment and more amenable to translating the setup to controlled in-situ studies. I would assume that an open system with fresh ultra zero air would just generate a concentration gradient and accelerate influx, such that the relative difference between actual and perceived gas concentration would be greater at high flow rates compared to low flow rates after accounting for dilution. But that there is also a counteracting equilibration time effect at fast flow rates (which is what is shown in figure 4 and 6, without necessarily parsing out the magnitude of concentration gradient and time to equilibrium effects).

Answer: For this initial study we wanted to evaluate the integration of the sintered PTFE soil probes and the new TILDAS with a reduced sample cell volume with a continuous method, which can be less complex for field deployment and less prone to create artifacts during sampling. Additionally, using the open flow-through method will allow the analysis of the same sample by other instruments connected in-line with TILDAS like PTR-MS Vocus-GC instruments. We kindly refer the reviewer to the previous response to clarify the equilibration time effect at fast flow rates.

RC2: Maybe something to speculate on whether it would be possible to make the detector volumes even smaller so that the probe volumes could be smaller and perhaps depth-resolved, or whether the whole system would need to be re-developed. This would be a really useful for any system where the soil is strongly depth resolved in

terms of chemistry and/or temperature (ex thawing tundra or forest floor).

Answer: We appreciate the reviewer's suggestion. We agree that decreasing the analyzer volume demand has been a challenge for soil trace gas detection in soils and the previous sampling systems using diffusive sample transfer. Laser spectrometers still require sufficient pathlength to quantify trace amounts of N2O isotopes, for example, so there is a trade-off between cell volume size and pathlength that we addressed here with the novel volume reducing insert. There is promise to reduce sample cells and pathlengths for some trace gases, especially when not determining isotopes. We are currently challenging the sampling system spatial resolution in both laboratory and field studies, which will be the topic of future papers.

RC2: Soil texture for S1: if the clay+silt = 66%, then the sand cannot be > 34%. L600 with rather than without?

Answer: We agree, the reviewer makes a good point. To avoid confusion, we will add the following phrase "predominant soil texture is sandy clay loam (20-35% clay-loam and remaining fraction sand " In table S1

---

## Author Comment (AC2) · 1 Apr 2021

Dear Dr. Nicolas Brüggemann and Reviewer #3,

We appreciate the detailed review and the honest comments from reviewer 3 to improve the manuscript. Following, we will address each of the reviewers comments and suggestions, those that required edits to the manuscripts will be addressed and will be noted in each response if pertinent for the editors' evaluation.

1. Does the paper address relevant scientific questions within the scope of BG? Without any doubt reliable protocols for diffusive soil-gas sampling combined with multi-component concentration and isotopic analysis are issues in soil soil ecological research. Indeed, progress in field research about soil-gas exchange requires reduced impact diffusive sampling with good temporal and spatial resolution. Another point are costs, that allow installation with sufficient spatial repetition. Therefore the topic of the paper adress clearly aims and scope of BG. My problem with the paper is, that in the study itself the critical issues are not treated critically and in comparison to existing solutions. The paper lacks a well defined scientific question. It seems to me more than an advertisement for a specific technical solution.

Answer: Thanks to the reviewer for honest comments and suggestions. We consider our research paper to be honest and open about the evaluation and optimization of an open-flow-through sampling system that is not widely used in the field. We have presented the challenges encountered during the evaluation of the system, and shared details of how they were solved. The sampling probes and sample transfer system are not yet commercially available. We hope this study is replicated by other researchers in this field to use and/or compare this probe and system with other instruments to address the challenges in BG noted by the reviewer. Our objective was to optimize and evaluate the versatility of the system under controlled conditions. To clarify the scientific question, we can highlight it in the introduction (in paragraph 6) to read as: "how can diffusive sampling approaches be used to obtain representative soil gas samples, with a return rate that allows high time resolution to capture changes in soil trace gases in response to manipulated environmental conditions." We will highlight also that by answering this question on improving and addressing the challenges of the current system, it will open the possibilities of future studies with sufficient spatiotemporal resolution to analyze soil trace gas dynamics in soil.

2. Does the paper present novel concepts, ideas, tools, or data? Neither diffusive gas sampling, nor the use of hydrophobic porous tubes are novel concepts. The authors use an obviously sophisticated assembly of gas analyzers covering a wide range of isotopic and concentration responses. However, I miss a comparative discussion about the novelty and the advantages of the used equipment. An interesting concept is presented in figure 12-b, where the isotopic signatures of N2O are plotted in a 3-D coordinate system and assigned to specific processes. However, this process analysis is not further discussed and the background of the interpretations are not reported with exception of a literature overview in the supplementary material.

Answer: We agree that the general concept of diffusive sampling approaches and hydrophobic porous tubes are not new. We focus on a novel material for the diffusive probes: sintered PTFE as the hydrophobic porous membrane instead of polypropylene or silicone. From personal experience by the authors and comparing with other experimental set-ups, we have found that sPTFE improves the hydrophobicity, biofouling and physical integrity of the soil probe. These advantages and comparisons were discussed in section "4.6 Implications for sPTFE as a field-based soil probe." However, we argue that the novelty of this submission is not in the probe alone, but its combination with other important developments in open-flow-through sampling scheme and analyzer volume reductions that allow us to sample soil trace gases using high-precision analyzers, with a return rate of minutes/hour for real-time measurements. Specifically, we consider that the novelty and advantages of the described system are the reduction of the sample volume demand of the instrument, ability to control online sample transfer and dilution from the same interface, and a reduction in sampling artifacts. These features allow the possibility of extended measurement periods with no need to transfer samples to the lab, and data acquisition for long periods of time (months/years) become possible. Additionally, the online, real-time and non-destructive sampling increases the possibility of analyzing a wide range of soil gases, including VOCs, isotopologues of various trace gases, including isotopically labeled compounds. Hence, this approach will increase possible experimental studies to understand microbial activity in the subsurface with better spatiotemporal resolution. We appreciate the reviewer's support for the N2O isotope concept presented in Figure 12b, which is inspired by work in pure

culture by the cited references. In this paper, we evaluate this system under controlled conditions to address our questions on whether the approach can sample representative soil gases and under which conditions. The goal of Figure 12b is to demonstrate one potential analysis path enabled by the new soil gas sampling system. As noted by the reviewer, the scope of these experiments already makes the paper quite lengthy, so it is beyond the scope to conduct a detailed scientific analysis of the observed shifts in N2O isotopes shown here.

3. Are substantial conclusions reached? No.

Answer: We kindly disagree. We were able to address the challenges of the soil gas sampling system, successfully obtaining representative samples from different probes measured consecutively, and were able to detect fluctuations in soil gas concentrations and isotopic signature as a response to environmental changes. We describe in detail a general approach to leverage precision gas analyzers for subsurface soil gas studies.

4. Are the scientific methods and assumptions valid and clearly outlined? The description of the sampling and analysis system is difficult to read and full of technical details that do, however, not focus on critical issues. I clearly miss a list of the requirements that are expected from "a versatile soil-gas concentration and isotope monitoring". Insofar the reader does not really learn, what the authors expect from their equipment with regard to quantification limits, system disturbance, usability in the field, or costs.

Answer: Thank you for the comments to improve the understanding of the manuscript. We consider the technical details important for readers who are interested in developing or improving the gas sampling system from soil subsurface. We consider that we were able to explain in "Results" and "Discussion" the critical issues that we encountered during the evaluation of the sintered PTFE probes the overall sampling system, and the analyzer, including the time response, volume demand from the analyzer, sample equilibration, flow rates, return rate, among others. We will address the concerns about any specific critical issue that the reviewer thinks we have missed.

We agree with the reviewer about the need to highlight why we consider the system versatile for soil gas concentration and isotope monitoring. In particular, this investigation showed that 1) the system has the potential to be used with other gas and isotope analyzers, 2) there was no evidence of any interference during the TILDAS-PTR-MS Vocus inline measurements, and 3) the nitrous oxide analyzer configuration achieved a reduced concentration dependency allowing determination of N2O isotopic measurements over a larger range in concentration. And importantly, the sampling and analysis system is able to capture fluctuations in subsurface gas concentration and isotopologues in response to fairly rapid changes in environmental conditions, with good sampling resolution in space and time.

The reader can find the quantification limits of the instrument in the supplement information on figures S3, S4, S2. These figures are annotated in the manuscript, however, for clarification we can highlight this point directly in the manuscript. Regarding the usability of field deployment, in section "4.7 Considerations for field deployment of the system" and "Conclusions", we stated several considerations for field deployment. This study has been scaled-up, as mentioned previously, to an enclosed ecosystem to measure N2O, CH4 and VOCs from the surface and at different depths, we will share the challenges and considerations based on this experience in its pertinent publication. Additionally, in the future we plan to deploy the system in an agricultural field, which will allow us to share in future publications the challenges faced during a field application.

5. Are the results sufficient to support the interpretations and conclusions? I miss specific scientific interpretations and conclusions with regard to a reference. The extensive report of results is somewhat anecdotal.

Answer: The references we used for N2O and CH4 concentrations and isotopologues are described in the last paragraph in the section "2.2.2 Novel laser spectrometer for N2O and CH4 isotopomers". Also the results were discussed based on dilution corrected and calibrated measurements.

6. Is the description of experiments and calculations sufficiently complete and precise to allow their reproduction by fellow scientists (traceability of results)? The technical descriptions are very detailed but not in such way, that it could be used to build up a system for measurement in soil columns or even in the field. One of several issues not adressed are power requirements and external conditions to work with the assembly of gas analyzers under field conditions. Neither I found suggestions for low-impact installation of the novel gas samplers or dead volume of the system.. So I have to answer with no.

Answer: We kindly disagree that the system description is not detailed in a way for others to replicate; the diagrams, component information, vendors and part numbers, and operational parameters are given with a level of detail comparable to similar reports, and we would argue with more detail in many cases. If the reviewer has suggestions for experimental details that are necessary to reproduce the system elsewhere, we will consider them. We explicitly describe how the system is used to test specific research questions in soil columns in the lab. Demonstrating a field deployment of the system is beyond the scope of this paper, but we do give specific recommendations for consideration in section 4.7 in the paper. To our knowledge, and in our prior experience publishing on ecosystem- and soil-flux measurement systems, it is not standard practice to list power requirements of sampling systems. As we do not demonstrate field sampling here, we don't list specific field instrument housing or sampling condition requirements, but these are well documented for the analyzers used here (Dual Tracer Gas Monitor, Aerodyne Research, Inc). We also see great potential to integrate lower-cost laser spectrometers and other monitors with these soil gas sampling probes, many of which already have smaller sampling volumes. Here we focus on integration with high sensitivity trace gas analyzers that allow additional compounds to be measured, which are more challenging given flow/volume demands.

We thank the reviewer for the good suggestion to provide recommendations for low-impact installation of soil probes to reduce impacts for dead volume. We can expand

the "4.7. Considerations for field deployment" section by discussing that a robust material like sintered PTFE will have the potential to adapt to deployment with devices that will facilitate direct installation in the subsurface, instead of a pre-drilled hole. However, the scope of this study was to evaluate the integration of the system in a controlled environment in the lab; therefore, we placed probes in the empty columns before burying, resulting in this being a non-issue in these experiments. In future studies we will be testing different installation methods in the field.

7. Do the authors give proper credit to related work and clearly indicate their own new/original contribution? The paper is more a technical description or even an advertisement of a specific setup but not mainly a scientific paper. It is not fully clear what is new (the use of PTFE instead of PE porous tubing?) or the use of an assembly of different laser-gas analyzers? Several literature references seem to more be an enumeration than a than a critical review.

Answer: We appreciate the author's comments to help us improve the paper by clarifying what we consider it has been an improvement in the soil gas analysis methods. As we have seen in other BG papers, we pose specific technical scientific questions that we answer with a series of controlled experiments, for example:

a. Pirk, Norbert, Mikhail Mastepanov, Frans-Jan W. Parmentier, Magnus Lund, Patrick Crill, and Torben R. Christensen. 2016. "Calculations of Automatic Chamber Flux Measurements of Methane and Carbon Dioxide Using Short Time Series of Concentrations." Biogeosciences 13 (4): 903–12.

b. Pape, L., Ammann, C., Nyfeler-Brunner, A., Spirig, C., Hens, K., and Meixner, F. X.: An automated dynamic chamber system for surface exchange measurement of non-reactive and reactive trace gases of grassland ecosystems, Biogeosciences, 6, 405–429, https://doi.org/10.5194/bg-6-405-2009, 2009.

We consider that the original contribution lays on the methodology evaluated and the improvements in the use of sintered PTFE material as a low reactive soil probe, innovation in the analyzer, precision in controlling sample transfer, and consecutive soil gas measurements from one sample with low volume demand. This work overcomes previous challenges to integrating the relatively high sample volume demand of many state-of-the-art trace gas analyzers (e.g., laser spectrometers but also online mass spectrometers) with diffusive soil probes for high resolution sampling both in space and time, instead of requiring significant compromise between the two. Therefore, the system required several modifications and changes during the evaluation process. Our approaches and elements of the system, including probe material, are not exactly the same as other systems in our literature references, and in this study we utilize trace gas instrumentation that has not previously been coupled to subsurface probes; therefore, we have not found a succinct and accessible way to compare directly with other soil gas sampling systems, as each system takes a different approach to address requirements of the analyzer and application. This resulted in the use of sintered PTFE that decreases reactivity of the material while being in contact with soil and changes in the soil environment like increase in moisture or biofouling, keeping its integrity and hydrophobicity with different matrices. Additionally, we consider this system is novel because it is the first online, realtime system quantifying subsurface soil gas N2O isotopes including site preference. The soil probes system soil has potential to be evaluated with low-cost analyzers, or to take high-precision analyzers to the field to measure a wide range of soil gases and isotopologues with a high spatiotemporal resolution. Respectfully, we consider that this is not an intercomparison paper, the strength of this paper is a sample transfer and probes approach that can be translated to other applications and a system that open the possibilities to evaluate in situ microbial activity from soils by measuring N2O isotopologues and a wide array of other trace gases including volatile organic compounds.

8. Does the title clearly reflect the contents of the paper? In principle yes.

9. Does the abstract provide a concise and complete summary? Yes, including the weaknesses of the scientific substance.

10. Is the overall presentation well-structured and clear? Too extensive, too complex, lacking structure and conciseness.

Answer: In organizing the manuscript, we took great care to structure the paper in specific, labelled sections/subsections and tables. In the resubmission, we will look for places where we can condense and clarify our presentation.

11. Is the language fluent and precise? Yes.

12. Are mathematical formulae, symbols, abbreviations, and units correctly defined and used? Yes

13. Should any parts of the paper (text, formulae, figures, tables) be clarified, reduced, combined, or eliminated? For publication the whole paper should be focused, shortened and reorganized including the formulation of a clear scientific question that can be discussed based on hypotheses. I would suggest to split up the paper into a physically based sampling optimization part and another, multi component (isotopic) analysis part and possibly the process study of the process interpretation by isotopic signatures.

Answer: We appreciate the suggestions to improve the paper. We can clarify the scientific question and the hypothesis. Our objective with this paper is to show the sampling optimization, as well as the versatility of the system. Therefore, we consider that the test with soil columns and isotopic analysis shows the potential of the system to study soil trace gas dynamic. Part of the process is to demonstrate the capabilities of this system in real soil to show what possibilities are available for others to replicate and apply in different, more complex settings. We also feel it is important to demonstrate the system in real soil so it is clear that it can be deployed in natural ecosystems.

14. Are the number and quality of references appropriate? Yes.

15. Is the amount and quality of supplementary material appropriate? Some parts of the supplementary material should be used in the main text. I miss in both, the paper and the supplementary material a listing of quantification limits of the equipment.

Answer: Thanks for the suggestions, we can highlight the precision and limitations in the text of the paper instead in the supplemental material. But here, we aim to balance the reviewer's concern for brevity vs adding more content to the main text. We checked that the precision and limitations are clearly referenced in the paper (Figures S3, S4, S2.). Or if the editor considers that the paper can be improved by moving figures and tables to the main paper, and specifies which content, we will do so.

---

## Author Comment (AC3) · 2 Apr 2021

Dear Dr. Nicolas Brüggemann,

We appreciate the constructive comments from reviewer 1 to improve the manuscript. This paper is the first of many steps to overcome the challenges of sample transfer on an online and real-time soil trace gas sampling system in soils. It highlights the need for innovating in soil probes, instrumentation, and sampling systems to detect soil trace gases in the subsurface with high sampling resolution.

Following, we will address each of the reviewer's comments and suggestions, those that required edits to the manuscripts will be addressed and will be noted in each response if pertinent for the editors' evaluation.

RC1: The importance of storage of gas in the air filled soil porosity on the link between soil concentration and net surface fluxes, and the dynamics of the air filled soil porosity, at short term (change in water content), long term (e.g. compaction), and also its spatial variation (e.g. local variation in bulk density), are not enough clearly stated and can be highlighted in the first paragraph of the introduction.

Answer: We agree with the reviewer's comment that the mentioned soil parameters are important in the soil trace gas dynamic. Based on the suggestion, we will modify the paragraph by adding the following (Line TBD): "Soil physical parameters play an important role in soil trace gas dynamics, where air-filled porosity influences soil trace gas concentrations and surface fluxes. Heterogeneity in soil structure and bulk density, changes in environmental conditions that alter soil water content (precipitation and temperature), and disturbances like soil compaction will change gas diffusivity and soil gas storage (Fletchard et al. 2007, Fujikawa and Miyazaki, 2005)."

Adding to bibliography: Fujikawa, Tomonori, and Tsuyoshi Miyazaki. "Effects of bulk density and soil type on the gas diffusion coefficient in repacked and undisturbed soils." Soil Science 170.11 (2005): 892-901

RC1: Tracing the fate of labeled gas in soil profile is also another promising approach that can be achieved by the development of such integrated monitoring tools. This can be mentioned in the second paragraph of the introduction.

Answer: Thanks to the reviewer for the suggestion and we agree that labeled stable isotopes and other gaseous tracers are interesting and promising tools that could be used with this integrated system to monitor and quantify consumption and production in soil.

Following the reviewer suggestion, we will add the following phrase to the second paragraph in the introduction paragraph to read as (Line TBD): "Previous studies used labeled 15N-N2O gas to determine consumption and production of N2O in soil columns (Clough et al., 2006) and inert helium gas as a tracer to quantify in-situ real-time measurement of gas transport in the soil at different depths (Laemmel et al., 2017). Subsurface isotopic tracer approaches will benefit from the development of an improved integrated systems that allow online quantification of trace gases in the soil as demonstrated by Laemmel et al., 2017."

Citations that would be included: Clough, T. J., et al. "Diffusion of 15N-labelled N2O into soil columns: a promising method to examine the fate of N2O in subsoils." Soil Biology and Biochemistry 38.6 (2006): 1462-1468.

Laemmel, T., et al. "An in situ method for real time measurement of gas transport in soil." European Journal of Soil Science 68.2 (2017): 156-166.

RC1: The carrier gas (Ultra Zero Air) contains O2 which diffuses to the soil. Is there a risk that microbial processes under study are altered by this change in O2 concentration in the soil? Would it be better to use pure N2 instead? Would it be possible to use a closed loop system rather than an open one? Can you discuss your choice for an open-one?

Answer: The reviewer makes good points about the sampling system. In this study, we used UZA as a carrier gas, but we recognize O2 diffusing from the carrier stream during sampling could alter the soil gas environment if it was not already oxic, and we chose not to use N2 that might affect the nitrogen isotopes we were measuring. We can address this suggestion by mentioning that the following should be considered in future studies: 1) biogeochemical implications of adding substrates, UZA in this study, to the subsurface and need to test inert carrier gases like He; and 2) criteria for recirculating sampling approaches to outweigh flow-through approaches for a given application. Regarding the flow scheme, we decided to evaluate an open flow-through scheme to

obtain an equilibrated gas sample directly from the probe. The return rate of the multiprobe system allows time for probe equilibration under the appropriate flow schemes before the sample is sent to the analyzer. The probe gas concentration reaches a steady-state value, that we characterize in the paper. To promote diffusive equilibration and also minimize advection across the flow, it is important to maintain constant, controlled, and matched flow rates. In closed-loop sampling, it is more difficult to control return flow without pressure perturbations from the inline pump, but it is feasible. In addition to decreasing cell volume demand by adding an insert, we reduced the TILDAS sample volume demand by adding in-situ dilution, which also helped to reduce concentration dependency and possible condensation in the sample lines. However, dilution in closed-loop systems would prevent the sample from achieving equilibration if diluent is added upon each sample pass, mitigating the inherent value of a closed-loop approach. For analyzers that have low volume demand and limited concentration dependence, we agree that closed-loop sampling is a great approach, e.g., GC analysis (Laemmel et al., 2017, https://doi.org/10.1111/ejss.12412), or additionally for sampling with low-cost sensors, e.g., for CO2 (UMS CO2 Multichannel Monitoring-System with BGLD-300/BGLD-30 SOIL GAS LANCE described in Jochheim et al, (2018) https://doi.org/10.1002/jpln.201700259).

A closed-loop scheme that requires simultaneous, continuous flow through all probes can make the system more complicated, costly, and complex for field deployment. Further, some analyzers (e.g., mass spectrometers) are destructive (PTR-MS ionizes molecules for analysis), preventing the sampling loop from being circulated. However, other soil gas sampling methods (e.g., online GC and low-cost sensors) using a closed-loop system continues to be promising approaches to decrease the impact on gas composition and chemistry during subsurface gas sampling. If the editor considers that these sentences will clarify our choice for an open-loop and highlight the need to continue improving soil gas sampling systems, we will add to the manuscript.

RC1: The control gas that were used for the test using silica filled column looks a bit

low compared to expected soil concentration. Would the results of the test be different with CO2 or CH4 concentration of 1% or more?

Answer: Diffusion across the soil probe membrane should be directly proportional to the concentration gradient. With ultra zero air as the carrier gas, and thus an end member, we would expect that higher soil gas concentrations than in the silica tests would move more quickly across the membrane and may equilibrate faster. As we show in Fig. 11, the probe flow rate is a controlling factor on probe equilibration in real soil, both for N2O and CH4, and that there is a plateau in equilibration at low flow rates (<20-30 sccm) in both silica and real soil. This is a very good suggestion for future work to test even higher controlled soil gas concentrations in the silica matrix.

RC1: Can you elaborate a bit more and give a definition of "backgrounds" (L257-261)

Answer: We agree with the reviewer, this clarification will increase understanding of the term, we will extend the definition on section " 2.2.2 Novel laser spectrometer for N2O and CH4 isotopomers" to read as follows: "spectral backgrounds were recorded using the instrument auto-backgrounding routine. A sample spectrum is recorded with the instrument sample cell filled with UZA. This spectrum is used to normalize sample spectra, improving accuracy and sensitivity by accounting for changing instrument conditions and possible drift."

RC1: You confirm that lower probe sampling flow rates allow more time to equilibrate than do high flow rates. Would it be possible to use the initial 'pulse' of high gas concentrations at the beginning of the measurement with high flow rates until the steady-state value is reached to recompute the initial concentration in the soil air (peak integration)?

Answer: The reviewer brings up a good point. We have considered integrating the initial pulse of gas concentrations to quantify fully equilibrated samples and overcome the partial equilibration at steady-state. Attempting to do this on flow-through data could be complicated because the full pulse peak is not observed due to the slow TILDAS turnover time, but drops only to the equilibration concentration. We have been

experimenting with flow schemes to transfer only the 'plug' of soil gas inside the probe, backed by the UZA carrier. However, evaluating that approach alongside the flow-through approach described here would be out of the scope of an already long paper. Thus, we describe it as a potential next step in Section 4.2.

RC1: Section 3 title is "results and discussion" but section 4 is "discussion". Should section 3 be "results" only? Sometimes, delta is not well converted in the pdf and become a square instead

We thank the reviewer for the suggestion, we will change the typo error and delete "discussion" from Section 3 to reflect that it is "Results" only. Additionally, we will communicate with the editor to fix the incompatibility of the delta notation.

---

## Author Response (AR1)

June16th, 2021

**Re: Response to Manuscript ID: bg-2020-401**

Dear Dr. Nicolas Brüggemann,

We appreciate the reviewers' constructive comments as well as yours. We have carefully addressed each of the reviewers' comments and suggestions in the revised manuscript and believe these edits have improved the manuscript.

Based on the requirements for revised paper submissions, we are including the following documents with the appropriate changes based on reviewers' comments:

- Response to Reviewer and editor taking into account the new page, paragraph, and line number.

- Revised manuscript and Figure Source Files for Resubmission

In order to improve the manuscript as suggested we made the following modifications:

- We added text to address the comments while condensing the manuscript and ending on a balance of fewer words.
- We clarified the questions and hypotheses that motivate this study, highlighting the soil processes studied and their relevance to the soil-atmosphere interactions
- We more clearly distinguished our technical aims and achievements from our process-based hypotheses and results.

Below, please find our response to the fourth reviewers and detailed responses to the editor's comments and the actions we have taken to address each comment. Editor's and reviewer's comments are noted in italic Verdana font, our responses are in Time New Roman font in blue.

Sincerely,

Dr. Laura Meredith
Corresponding author

*commentbg-2020-40 1 Review by reviewer #4, Albrecht Neftel*

*I read with interest this paper and was impressed by the analytical development presented to characterize trace gas composition in the open pore space of a soil matrix. It is a pleasure to see a follow up of the membrane tube technique (METT) that we have developed many years ago.*
*The presented instrumental setup offers the potential to explore the large variability of microbial and chemical processes in soil that controls trace gas exchange within the soil and between the atmosphere and the soil. It is a milestone to get simultaneously access to continuous data on isotopic ratios of nitrous oxide (δ18O, δ15N, and the 15N site-preference of N2O), methane, carbon dioxide (δ13C), and VOCs.*
*The paper first presents data from a control experiment from an artificial inert soil imitation to characterize collection efficiency and reproducibility as the gas probing relies on passive diffusion through the porous membrane tube and obviously the gas flow will have a key influence on the measured concentrations.*

*Secondly data from packed soil core with an embedded sampling tube are presented. An N2O pulse as consequence of an irrigation was traced. The information on the isotopic signature of the N2O concentration in the soil allows to disentangle different production pathways for N2O. This is a valuable information as I most cases the interpretation of the mechanisms leading to an observed N2O flux is a lot of guessing.*

*We have been aware when we developed the METT system and analyzed the data, that in the best case we got representative trace gas concentrations (at that time we focused mainly on N2O and CO2) in an additional large pore artificially introduced in the soil. These concentrations might not be representative for the most important processes that control the N2O production and consumption as the oxygen concentration is likely higher as in small pores.*

*I have only a small criticism. The idea of articles in BG is on aspects of the interactions between the biological, chemical, and physical processes in terrestrial life with the geosphere, hydrosphere, and atmosphere. The paper has a very technical focus and presents a toll box what can be measured. The paper would gain in strength if a proposition what relevant question linking the different spheres would be given. I am perfectly aware that this is to moan on a high-level.*

*Albrecht Neftel*

*Neftel Research Expertise, Wohlen b Bern, Switzerland*

Dr. Neftel,

The authors appreciate your positive comments and the value you've found in our efforts to use isotopic signatures of soil trace gases as real time biomarkers of microbial metabolism, and keep working on improving soil gas sampling methodologies, which build from your work.

Based on your constructive suggestions, we have condensed and focused the paper to better highlight the questions, hypotheses and processes that we studied with our system and how they relate to biogeochemical interactions between the spheres.

*bg-2020-401 Comments by Associate*

*Dear Authors,*

*Your paper presents interesting aspects of a non-destructive technique for soil gas monitoring and isotope analysis. I found your paper well written and informative. Two of the reviewers also had a similar impression, while reviewer #3 was more critical. Reviewer #3 recommended "the whole paper should be focused, shortened and reorganized including t**he formulation of a clear scientific question that can be discussed based on hypotheses.** I would suggest splitting up the paper into a physically based sampling optimization part and another, multi component (isotopic) analysis part and possibly th**e process study of the process interpretation by isotopic signatures."** Please follow this recommendation as far as possible.*

*In the meantime, I received the review of a fourth reviewer, which could not be uploaded due to technical reasons (included in the attached file). But he also approves your paper and had only a few minor comments.*

*Finally, I also went through the paper in detail and had only a few comments and technical corrections (see attached file).*

*Considering all the comments and your responses, I have decided to recommend your paper for publication after major revisions. Please address all comments and provide a point-by-point response.*

*General comments*

Following the editors and reviewers comments, and trying to make the paper more succinct, we: i) we highlighted the questions and hypotheses in the introduction in the last paragraph of the introduction in page 4, and ii) the section from 4.1 to 4.4. Are dedicated to the integration and optimization process of the system, and we merged the discussion sections 4.4 and 4.5 into one called "Subsurface gas measurements to capture and interpret environmental drivers of soil processes" in page 29.

*Editor Specific comments:*

*p. 3, L. 72: „For example, probes larger than 1 m have been used in water": You cite Rothfuss et al. (2013) for this statement, but see Rothfuss et al. (2015), who used 15 cm long pieces of the same microporous PP tubing (Accurel) successfully for water isotope measurements over a period of 290 days.*

Thanks for the suggestion. Later in line 73, we mentioned the PP accurel improving the diffusion equilibrium time. In this line we meant to cite Rothfuss et al. (2015). Also we found value highlighting the specific probe and the sampling period. The sentence in Line 73 now will read: "Rothfuss et al. 2015 used a 15 cm PP tubing to measure water isotope for 290 days."

*p. 10, L. 221-222: You mention here that the TILDAS you used was also capable of measuring water isotopologues, but you don't present any data. For the readers who are interested in non-destructive analysis of soil water isotopic composition, it would be very interesting to see the performance of your soil probes also for soil water isotopic analysis.*

Our controlled optimization experiments were run using a dry silica matrix flushed with dry air from compressed gas tanks, so our soil moisture and water vapor concentrations were low, and not controlled in the experiments that used the water isotope analyzer. We agree with the Editor that this method will be interesting for the hydrological community, and hope to have time in the future to perform the controlled experiments with silica and soil to share useful data with the community.

*p. 11, L. 271: Here you mention a surveillance standard of 1,000 ppm N2O. From the following sections it can be deduced that it should read 1,000 ppb here. Please confirm.*

We are confirming that we used the MIT Ref II to make a 3 L surveillance standard of ~1000 ppm $N_2O$, which was then diluted into a container filled with zero air to produce sampled concentrations of 100 ppb - 30 ppm.

*p. 12, L. 284: Use capital delta here: m/Δm.*

Thanks for the correction, the line 278 now reads as suggested > 10000 m/Δm

*p. 15, L. 345: It is not clear why the 2196 cm-1 region was chosen for N2O isotopocules. There is a more suitable region between 2203-2203.4 cm-1, where all four N2O isotopologue lines are found at similar transmittance values between 0.9995 (weakest = 15N14N16O) to 0.998 (strongest = 14N14N16O). This would strongly reduce any issues with non-linearity (= concentration dependence). Although there is a relatively strong CO line at about 2203.16 cm-1, its interference at higher concentration can be reduced by removing the CO from the air stream (see Ibraim et al., 2018,*
*Isotopes in Environmental and Health Studies, 54(1), 1-15. Doi: 10.1080/10256016.2017.1345902*

Through simulations and experimental testing, we determined that the 2196cm-1 region is overall a better spectral region for monitoring the $N_2O$ isotopologues (i.e., 446, 456, 546, and 448) in the soil gas matrix. There are fewer interfering or closely overlapping lines, such as from $CO_2$ which is expected to be present on the percent level. A comparison of those two regions is shown below, with the same vertical scale. $CO_2$ at 2% concentration would make very strong interference with the 456 and 546 absorptions. The line strengths of the $N_2O$ species are also stronger at 2196cm-1 for the 15N isotopologues. While we agree that a CO scrubber could remove CO spectral interference, it would add complication to the flow scheme.

[Figure]

*p. 19, L. 404: What is shown in Figure 6 compared to Figure 4? The data look quite different, but it is not clear to me what was the difference in setup or measurement.*

Figure 4 shows the effect of probe sampling on the column by changing the probe flow rate with constant control gas concentration and dilution using System 1 and a single column. We alternated measurement of $CO_2$ concentration in headspace gas (1 h) and the probe (15 min) . The column $CO_2$ was depleted after probe sampling and took 1 hour to stabilize. To further clarify this, we added the following to the Figure 4 caption, line 369: ", representing the potential impact of probe sampling on the soil environment". In Figure 6, we evaluated the impact of different total flow rates and dilutions at different percent increments calculating the residence time explaining the dependence on flow probe rate. To clarify this, we added the following to the Figure 6 caption in line 394: ", reflecting the recovered sample vs. true gas concentrations, respectively".

*p. 20, L. 419-420: "These concentration and isotopic fractionation results underscore the need to ensure that the probe flow rate is sufficiently low…": Yes, or that the probe is sufficiently long (!) to allow a reasonably high gas flow required for the analyzers, especially at low soil gas concentrations where dilution would compromise the analyzer precision, especially for isotope measurements. This point is missing in the discussion, i.e. to ponder whether the shortness of the probes used bring also a disadvantage (= too strong a dilution of soil gas at higher sample flow rates through the probes), which could be overcome by longer probes.*

The Editor raises a good point. We mainly advocate for the use of shorter probes to reduce the sampling footprint of the probe and resolve a smaller region in the soil, but this clearly is a challenge to obtaining a well-equilibrated sample. We argue that in most cases smaller probes are advantageous for increasing spatial resolution and minimizing disruption, but we also recognize that a longer sampling length might integrate over the heterogeneity in soil and be considered an advantage for other applications. To address this, we added the following passage to the discussion on Page 29, Line 590: "In some field applications, it may be more desirable to physically integrate (rather than resolve) variations in soil gas concentrations over a distance (e.g., for a representative concentration) using a long soil probe, which would help release the low-flow demands of the relatively short probes used here."

*p. 24, L. 482: It would be good to have an estimate of the precision of your SP values, especially in view of the fact that it is the difference of two isotope ratios. Looking at your Figure 9, it seems as if the SP precision could easily be >10‰, making any strong statement on source processes basically impossible.*

The precision of the SP values (and any isotope measurement) is strongly dependent upon concentration. Therefore there are concentrations for which an SP value would be too noisy to be useful. Here, the measured 15Nbulk and SP precisions were 0.9‰ and 1.6‰, respectively (page 10, line 258), at 325 ppb with an averaging time of 2 minutes. Importantly, barring significant N2O consumption in soil, concentrations are typically larger, from 1-100 ppm, putting the isotopic ratios well below 1‰.

*p. 29, L. 611: Also Gangi et al. (2015), mentioned in your reference list, used microporous PP tubing for soil CO₂ isotope measurements, and Rothfuss et al. (2013) and (2015) for soil water isotope analysis, without any problems regarding physical/mechanical stability or loss of hydrophobicity.*

Thank you for the recommendation and clarification, we added Gangi, et al, 2015 in our references for PP probes for $CO_2$ on Page 28, Line 583. Additionally, we will highlight the successful use of PP for water isotopes on Page 28, Line 585 to read: "PP has been successfully used for water isotope analysis (Rothfuss et al. 2013; Rothfuss et al. 2015).."

*P. 30, L. 621: From your work, it did not become clear how large the soil volume is that is affected by the probe, which ultimately determines the (reasonable) spatial resolution. This should be taken into account here when talking about cm-level spatial resolution.*

From our controlled tests, we are not able to determine whether the impact of probe sampling on soil gas concentrations was a big effect on a small volume around the probe, or a dilute effect over a large volume. This is especially complicated with the controlled tests where we flow control gas through the column, and replenish controlled gas around the probe faster than we would expect in the field. We believe the best test for this would be to install multiple probes in a column to evaluate the reach of probe sampling, which we aim to do in future tests. Here, we could do a back of the envelope calculation:

The gas resides in pore space, so whatever pores are not filled with water carry the soil gas that diffuses into the probe. Typical soil porosities are 40-50%. If we fully exchange, say 20 mL of soil gas for sampling from dry soil, the actual soil volume sampled by the probe is then 20 ml/45% = 44 mL.  Note that this is soil moisture-dependent. For wet soil, if the water filled pore space (WFPS) fraction is, e.g. 75%, then the volume of available pore spaces is 75% less, and the footprint of the measurement would increase 4-fold to 176 mL. That value is an upper limit, however, because the water in the pores has the soil gas dissolved in it as well (in equilibrium). To take that into account when calculating the footprint would require knowledge of the Henry's or Raoult's Law coefficients for the analyte species.

For probe dimensions of 1.25 cm diameter and 15 cm long, 44 mL of soil volume corresponds to a region of soil extending ~0.5 cm away from the probe surface. 176 mL corresponds to a region of soil extending ~1.4 cm away from the probe surface.

***Technical corrections:***

*p. 2, L. 38: VOC was defined already in L. 35 on the same page.*

It was corrected in Page 1, line 40 to read: "...and VOCs (Abis et al., 2020; Raza et al., 2017)"

*p. 15, L. 349: Figure 3, caption: a) and b) have been scrambled and need to be swapped.*

The order of the plots (page 15) in the figure were swapped to match the order of the reference in the manuscript.

*p. 18, L. 393: Figure 5: What is the unit of time? I assume minutes,*

Time units were added to the figure 5, now it is on Pag 17.

*please add. p. 19, L. 410: Change 20C to 20°C.*

The symbol for centigrade degrees was added to read 20°C in page 18, line 400.

*p. 26, L. 514: "a few hours delay"*

Thanks to the editor for noticing the mistake. The phrase in page 25, line 513 was changed to read: "In contrast, after five hours,..."

*p. 26, L. 515: The formula of dimethyl sulfide must read either $C_2H_6S$ or $(CH_3)_2S$.*

Thanks for noticing the mistake, the formula was fixed to read $C_2H_6S$ in page 25, line 514.

---

## Editor Decision (ED1)

bg-2020-401 Review by reviewer #4, Albrecht Neftel

I read with interest this paper and was impressed by the analytical development presented to characterize trace gas composition in the open pore space of a soil matrix. It is a pleasure to see a follow up of the membrane tube technique (METT) that we have developed many years ago.
The presented instrumental setup offers the potential to explore the large variability of microbial and chemical processes in soil that controls trace gas exchange within the soil ad between the atmosphere and the soil. It is a milestone to get simultaneously access to continuous data on isotopic ratios of nitrous oxide ($\delta18O$, $\delta15N$, and the 21 15N site-preference of N2O), methane, carbon dioxide ($\delta13$ 22 C), and VOCs.
The paper first presents data from a control experiment from an artificial inert soil imitation to characterize collection efficiency and reproducibility as the gas probing relies on passive diffusion through the porous membrane tube and obviously the gas flow will have a key influence on the measured concentrations.
Secondly data from packed soil core with an embedded sampling tube are presented. An N2O pulse as consequence of an irrigation was traced. The information on the isotopic signature of the N2O concentration in the soil allows to disentangle different production pathways for N2O. This is a valuable information as I most cases the interpretation of the mechanisms leading to an observed N2O flux is a lot of guessing.
We have been aware when we developed the METT system and analyzed the data, that in the best case we got representative trace gas concentrations (at that time we focused mainly on N2O and CO2) in an additional large pore artificially introduced in the soil. These concentrations might not be representative for the most important processes that control the N2O production and consumption as the oxygen concentration is likely higher as in small pores.
I have only a small criticism. The idea of articles in BG is on aspects of the interactions between the biological, chemical, and physical processes in terrestrial life with the geosphere, hydrosphere, and atmosphere. The paper has a very technical focus and presents a toll box what can be measured. The paper would gain in strength if a proposition what relevant question linking the different spheres would be given. I am perfectly aware that this is to moan on a high-level.

Albrecht Neftel
Neftel Research Expertise, Wohlen b Bern, Switzerland

bg-2020-401 Comments by Associate Editor

Specific comments:

p. 3, L. 72: „For example, probes larger than 1 m have been used in water": You cite Rothfuss et al. (2013) for this statement, but see Rothfuss et al. (2015), who used 15 cm long pieces of the same microporous PP tubing (Accurel) successfully for water isotope measurements over a period of 290 days.

p. 10, L. 221-222: You mention here that the TILDAS you used was also capable of measuring water isotopologues, but you don't present any data. For the readers who are interested in non-destructive analysis of soil water isotopic composition, it would be very interesting to see the performance of your soil probes also for soil water isotopic analysis.

p. 11, L. 271: Here you mention a surveillance standard of 1,000 ppm N2O. From the following sections it can be deduced that it should read 1,000 ppb here. Please confirm.

p. 12, L. 284: Use capital delta here: m/Δm.

p. 15, L. 345: It is not clear why the 2196 cm-1 region was chosen for N2O isotopocules. There is a more suitable region between 2203-2203.4 cm-1, where all four $N_2O$ isotopologue lines are found at similar transmittance values between 0.9995 (weakest = 15N14N16O) to 0.998 (strongest = 14N14N16O). This would strongly reduce any issues with non-linearity (= concentration dependence). Although there is a relatively strong CO line at about 2203.16 cm-1, its interference at higher concentration can be reduced by removing the CO from the air stream (see Ibraim et al., 2018, Isotopes in Environmental and Health Studies, 54(1), 1-15. Doi: 10.1080/10256016.2017.1345902

p. 19, L. 404: What is shown in Figure 6 compared to Figure 4? The data look quite different, but it is not clear to me what was the difference in setup or measurement.

p. 20, L. 419-420: "These concentration and isotopic fractionation results underscore the need to ensure that the probe flow rate is sufficiently low…": Yes, or that the probe is sufficiently long (!) to allow a reasonably high gas flow required for the analyzers, especially at low soil gas concentrations where dilution would compromise the analyzer precision, especially for isotope measurements. This point is missing in the discussion, i.e. to ponder whether the shortness of the probes used bring also a disadvantage (= too strong a dilution of soil gas at higher sample flow rates through the probes), which could be overcome by longer probes.

p. 24, L. 482: It would be good to have an estimate of the precision of your SP values, especially in view of the fact that it is the difference of two isotope ratios. Looking at your Figure 9, it seems as if the SP precision could easily be >10‰, making any strong statement on source processes basically impossible.

p. 29, L. 611: Also Gangi et al. (2015), mentioned in your reference list, used microporous PP tubing for soil $CO_2$ isotope measurements, and Rothfuss et al. (2013) and (2015) for soil water isotope analysis, without any problems regarding physical/mechanical stability or loss of hydrophobicity.

P. 30, L. 621: From your work, it did not become clear how large the soil volume is that is affected by the probe, which ultimately determines the (reasonable) spatial resolution. This should be taken into account here when talking about cm-level spatial resolution.

Technical corrections:

p. 2, L. 38: VOC was defined already in L. 35 on the same page.

p. 15, L. 349: Figure 3, caption: a) and b) have been scrambled and need to be swapped.

p. 18, L. 393: Figure 5: What is the unit of time? I assume minutes, please add.

p. 19, L. 410: Change 20C to 20°C.

p. 26, L. 514: "a few hours delay"

p. 26, L. 515: The formula of dimethyl sulfide must read either $C_2H_6S$ or $(CH_3)_2S$.

---

## Author Response (AR2)

November 8th, 2021

**Re: Response to Manuscript ID: bg-2020-401 comments to resubmission**

Dear Dr. Brüggemann
Associate Editor Biogeoscience

Thank you very much for recommending our paper for publication. We appreciate your time and expertise to help us improve the paper.

Based on your latest comments and suggestions, we will address in Time New Roman font in blue to each of your suggestions (Italics-Verdana font).

Sincerely,

Dr. Laura Meredith
Corresponding author

**Comments to the author by Dr. Nicolas Brüggemann Associate Editor decision**

*p. 4, L117: I stumbled across "translatability for" (I am not a native speaker, though). Maybe "tranferability to" fits better here? Or "...approach that is robust, flexible, and transferable to..."?*

Dr. Brüggemann we used the word "translatability" to describe that information obtained by the integration of the system can be used to reflect or understand microbial activity in soil. However, we appreciate your comment and we changed the word to "transferability" to describe that the method can be used for other scientific questions while keeping the embedded value into another application.

*p. 10, L257: Please add the information on mixing ratio and averaging time to the precision values (e.g., for N2O you specified 325 ppb and 2 min in your response letter).*

Based on your suggestion we edited the phrase to read in p. 10, L257-258 "Measured instrumental precisions with an averaging time of 2 minutes were 0.9‰ and 1.6‰ for N2O bulk 15N and site preference, respectively, at 325 ppb N2O, and 0.2‰ for 13CH4

*p. 29, L625: "subsurface processing producing": it's probably "subsurface processes producing"*

Thank you very much for noticing this typo. The word "processing" was changed to "processes" as suggested now **in p.29 L622**